# Characteristics of the summer atmospheric boundary layer height over the Tibetan Plateau and influential factors

Junhui Che[1, 2, 3], Ping Zhao[1]

[1]State Key Laboratory of Severe Weather, Chinese Academy of Meteorological Sciences, Beijing, 100081, China

[2]Collaborative Innovation Center on Forecast and Evaluation of Meteorological Disasters, Nanjing University of Information Science and Technology, Nanjing 210044, China

[3]Shandong Meteorological Service Center, Jinan, 250031, China

*Correspondence to*: Ping Zhao (zhaop@cma.gov.cn)

**Abstract** The important roles of the Tibetan Plateau (TP) atmospheric boundary layer (ABL) in climate, weather and air quality have long been recognized, but little is known about the TP ABL climatological features and their west-east discrepancies due to the scarce data in the western TP. Based on intensive sounding, surface sensible heat flux, solar radiation, and soil moisture observational datasets from the Third Tibetan Plateau Atmospheric Scientific Experiment and the routine meteorological operational sounding and ground-based cloud cover datasets in the Tibetan Plateau for the period 2013-2015, we firstly investigate the west-east differences in summer ABL features over the TP and the associated influential factors. It is found that the heights of both the convective boundary layer (CBL) and the neutral boundary layer (NBL) exhibit a diurnal variation and a west-east difference in the TP, while these features are not remarkable for the stable boundary layer (SBL). Moreover, the ABL shows significant discrepancies in the amplitude of the diurnal variation and the persistent time of the development between the eastern and western TP. In the early morning (08:00 BJT), the ABL height distribution is narrow, with a mean height below 450 m above ground level (AGL) and a small west-east difference. The SBL observed at this moment accounts for 85% of the TP total ABL. There are a wide distribution in the ABL height up to 4000 m AGL and a large west-east difference for the total ABL height at noon (14:00 BJT), with a mean height above 2000 m AGL in the western TP and around 1500 m AGL in the eastern TP. The CBL accounts for 77% of the TP total ABL at this moment, with more than 50% of the CBL above 1900 m AGL. In the late afternoon (20:00 BJT), the CBL and SBL dominate the western and eastern TP, respectively, which results in a larger west-east difference of 1054.2 m between the western and eastern TP. The high ABL height in a cold environment over the western TP (relative to the plain areas) is similar to that in some extreme hot and arid areas such as Dunhuang and Taklimakan Deserts. In general, for the western (eastern) TP, there is low (high) total cloud coverage, with large (small) solar radiation at the surface and dry (wet) soil. These features lead to high (low) sensible heat flux and thus promotes (inhibits) the local ABL development. This study provides new insights for west-east structures of the summer ABL height, occurrence frequency, and diurnal amplitude over the TP region and the associated reasons.

## 1 Introduction

The atmospheric boundary layer (ABL) commonly refers to the bottom layer of the troposphere directly coupled with the earth's surface at a response time scale of about one hour or less, in which a variety of complex motions characterized by turbulence may be present (Stull, 1988). The turbulent motions in the ABL are responsible for the atmospheric mixing processes, which affects the vertical redistribution of water vapour, momentum, heat, and atmospheric pollutants (Stull, 1988; Garratt, 1992; Huang et al., 2007; Miao et al., 2015). The ABL height (ABLH) as a fundamental variable is critical to diagnose turbulent mixing, vertical disturbance, convective transport, pollutant dispersion, and atmospheric environmental and effective heat capacity (Garratt, 1993; Seibert et al., 2000; Guo et al., 2009; Esau and Zilitinkevich, 2010; Dai et al., 2014; Pal and Haeffelin, 2015; Davy and Esau, 2016). Therefore, the accurate specification of the ABLH is essential to develop weather, climate, and air pollution prediction models.

The cloud-free ABL overland can be divided into three types, that is, the convective boundary layer (CBL), the stable boundary layer (SBL), and the neutral boundary layer (NBL) (Stull, 1988). The CBL usually has the strongest turbulence forced by surface buoyancy flux with or without wind shear, and is generally capped by a strong temperature inversion maintained through large-scale subsidence. The CBL height is a result of the balance of the turbulence-induced entrainment and the subsidence velocity (e.g. Driedonks and Tennekes, 1984). However, turbulence in the SBL is mainly driven by the mean wind shear against negative buoyancy flux from the stable thermal stratification within the nocturnal surface inversion (NSI). The SBL height is hence related to the boundary layer wind and wind shear, which sometimes are used to identify the SBL height. The NBL occurs in neutral conditions with the turbulence of almost the same intensity in all directions (Stull, 1988; Blay-Carreras et al., 2014). It denotes the type of boundary layer with solely wind forcing and normally occurs during the transition from the daytime CBL to the night time SBL. It can also occur anytime when the buoyancy forcing is weak. The ABLH variability is dominated by its strong diurnal cycle (Stull, 1988; Garratt, 1992). In this diurnal cycle, the different manifestations of an ABLH are generated in response to the distinct forcing mechanisms that originate from mechanical (wind shear) and thermal (buoyancy) effects (Stull, 1988; Garratt, 1992). Over land and after sunrise, the surface is heated by solar radiation, resulting in upward heat flux that initiates strong updrafts of warm air. Such a mechanism generates a deepening of the CBL (Chen and Houze, 1997). At sunset, the surface cools more rapidly compared to the air above, resulting in negative heat flux that consumes turbulent kinetic energy. Consequently, shear-driven turbulence can only maintain a shallow SBL with the formation of the NSI (Zhang et al., 2011a; Miao et al., 2015). Above the NSI, the convective energy-containing eddies start to lose their strength and mixing capacity. This deep and near-adiabatic vertical region, which is the remnant of the daytime CBL, is known as the residual layer (RL). The use of precise information on the RL in numerical models is of fundamental importance when describing the evolution of the diurnal CBL (Blay-Carreras et al., 2014; Chen et al., 2016).

The ABLH can be calculated from temperature, humidity, and wind profiles (Holtslag and Boville, 1993; Seibert et al., 2000;
Seidel et al., 2010; Bosveld et al., 2014; Davy, 2018). The CBL height is generally less than 2000–3000 m AGL and the SBL
thickness is usually less than 400–500 m AGL (Garratt, 1992). The ABL height shows an obvious spatial variation due to
differences in topography, thermal properties of the underlying surface, and weather conditions. For example, the CBL can
grow to the height of 4700 m AGL in New Delhi before the outbreak of the South Asian monsoon, whereas it only reaches
900 m AGL in Bangalore during the monsoon period (Raman et al., 1990). Seidel et al. (2010, 2012) pointed out that a large
east-west spatial gradient of the ABLH at sunset in the United States spanning several time zones may be conflated with the
diurnal variations of the ABL for the local solar time in the west earlier than in the east at fixed observation times. Guo et al.
(2016) identified three large-scale ABLH spatial patterns in China, that is, a west-east gradient during sunrise, an east-west
gradient during sunset, and a south-north gradient at noon. The reasons for the first two patterns are similar to those in the
United States shown in Seidel et al. (2012), while the south-north gradient may be related to the local surface and
hydrological processes (Guo et al., 2016; Zhang et al., 2017).

The Tibetan Plateau (TP) with an average elevation exceeding 4000 m is characterized by complex land surface processes
and boundary layer structures (Tao and Ding, 1981; Yanai and Li, 1994; Xu et al., 2002; Yang et al., 2004; Li and Gao, 2007;
Sun et al., 2007; Zhao et al., 2019b). The ABLH in the TP can reach 2000–3000 m AGL, generally higher compared to some
plains areas (with the ABLH of 1000–1500 m AGL) (Ye and Gao, 1979; Zhao and Miao, 1992; Xu et al., 2002; Zhang et al.,
2003). The ABLH in the TP varies greatly with location and season. At Gaize station of the western TP,the super-thick
ABLH may exceed 5000 m AGL during winter (Chen et al., 2013, 2016). In the central TP, the ABLH is lower, between 400
and 1800 m AGL at Dangxiong station and 1750 m AGL at Namucuo Lake (Li et al., 2000; Liu et al., 2001, and Lü et al.,
2008). Moreover, there is also a significant difference in the TP ABLH between dry and rainy seasons (Zuo et al., 2004). For
instance, the ABLH at Naqu station is 2211–4430 m AGL in the dry season, while it is 1006–2212 m AGL in the rainy
season (Li et al., 2011).

Although observations and studies for the TP ABL features have made progress, routine meteorological operational
sounding observations are scarce in the western TP due to the local high elevations, naturally harsh environmental conditions,
and logistic challenges. The previous studies on the ABL in the western TP are usually based on observational data at
Shiquanhe (during 15 days in one summer) and Gaize (during 22 days in one summer) stations (Song et al., 1984; Chen et al.,
2013). Thus, the statistical representation of their results is limited. Moreover, there are significant differences in surface
properties and general climate between the eastern and western TP (Wang et al., 2016). Few studies examined the west-east
differences in the ABL features due to the scarce data in the western TP. To obtain a longer observational data in the western
TP, the Third Tibetan Plateau Atmospheric Scientific Experiment (TIPEX-III) has made routine sounding launches at
Shiquanhe, Gaize, and Shenzha stations of the western TP (Fig. 1) since 2013, which fills in the data gaps in the operational
sounding network over the western TP (Zhao et al., 2018). Meanwhile, the TIPEX-III also carried out the intensive sounding
observations in the TP and adjacent stations at 14:00 Beijing Time (06:00 UTC) in June, July, and August (Zhao et al., 2018).
Compared to the previous field experiments over the TP, the TIPEX-III has a wider and longer coverage of sounding
observations over the western TP, providing valuable observational data for studying the ABL features in the western TP and
the west-east differences of these features in the TP during summer.

This study utilizes the TIPEX-III sounding observational data to investigate the features of the ABLH in the TP and their
differences between the western and eastern TP during summer, and analyzes the major factors affecting the ABLH in the
TP. The remainder of this paper is organized as follows. Main features of data and methods are described in Section 2. In
Section 3, the characteristics of the ABLH in the eastern and western TP and their regional differences are analyzed in detail.
In Section 4, the major factors affecting the ABLH in the TP and the west-east differences are examined. Discussions and
conclusions are given in Section 5.

## 2 Data and analysis methods

### 2.1 Observation data

The TIPEX-III carried out the intensive routine meteorological sounding observations at Shiquanhe (SQH), Gaize (GZ), and
Shenza (SZ) stations of the western TP (marked by red dots in Fig. 1) since the 2013 summer (Zhao et al., 2018), which have
been applied in research on the vertical structure of the upper troposphere and lower stratosphere at Gaize station during the
rainy season and the effects of assimilating the intensive sounding data on downstream rainfall (Hong et al., 2016; Yu et al.,
2018; Zhao et al., 2018; Zhao et al., 2019b). These intensive sounding data and the routine meteorological operational
sounding data at 16 stations of the central-eastern TP from the China Meteorological Administration (marked by black dots
in Fig. 1) are utilized in this study. The sounding observations at the above intensive and operational sounding stations were
carried out at 08:00 Beijing Time (BJT; 00:00 UTC), 14:00 BJT (06:00 UTC), and 20:00 BJT (12:00 UTC) each day during
summer (June, July, and August), including vertical profiles of temperature, humidity, and wind direction and speed. After
the quality control of the sounding observational data, we select data from three time periods for this study: 15 June to 31
July 2013, 15 June to 31 August 2014, and 1 June to 31 August 2015. There are 11,635 sounding profiles (Fig. 1a) from 19
stations over the TP region consisting of 4745, 2049, and 4841 profiles at 08:00 BJT (Fig. 1b), 14:00 BJT (Fig. 1c), and
20:00 BJT (Fig. 1d), respectively. It is evident that the observational sample size used in this study is much more compared
to the previous studies. Meanwhile, it is noted that there is a large difference in the sample size between the intensive and
operational observation records at 08:00 BJT and 20:00 BJT (Fig. 1b and d), which is called the original dataset for
convenience. Consequently, we also select the test group dataset which contains the same intensive observation records as
the operational ones at these two times to make sensitivity analysis (shown in Section 3.2), which shows that the difference
in the sample size between the intensive and operational observation records does not change our conclusions.

To analyze the factors affecting the ABL in the TP, we use the TIPEX-III 30-min mean surface sensible heat flux (SHF),
downward solar radiation, and 5-cm soil volume moisture content at SQH (bare soil with less vegetation), Naqu (NQ; alpine
steppe), and Linzhi (LZ; alpine meadow with few shrubs and trees) stations in the 2014-2015 summers (Wang et al., 2016;
Zhao et al., 2018; Li et al., 2019, 2020). In addition, the manual operational ground-based cloud cover observations at 02:00,
08:00, 14:00, and 20:00 BJT from the China Meteorological Administration are also used in this study. These ground-based
could cover data have been utilized by Guo et al. (2016) and Zhang et al. (2017).

**2.2 Calculation method of ABLH**

The potential temperature gradient method, proposed by Liu and Liang (2010) and sketched in Fig. 2a, is utilized in
identifying the ABL type and calculating the ABL height. The CBL height is defined at the base of the overlying inversion
layer that caps the rising convective thermals. The SBL height is defined as the top of the underlying inversion layer, where
turbulence decreasing from the surface nearly ceases (Stull 1988). In the evening and morning transition periods when the
RL may occur, the neutral RL starting from the surface is identified with near-neutral conditions in the surface layer (that is
the NBL).Following Liu and Liang (2010), Zhang et al. (2017), and Zhao et al. (2019a), the original sounding observation
profiles with a fine vertical resolution of ~1 hPa are interpolated to a vertical resolution of 5 hPa (corresponding to a
vertical interval around 50 m in the ABL) by the nearest neighbor interpolation method. On the basis of the near-surface
thermal gradient, such as a potential temperature ($\theta$) difference (*PTD*) between the fifth layer (~250 m; $\theta_5$) and the second
layer (~50 m; $\theta_2$), the ABL is classified as follows.
$$PTD = \theta_5 - \theta_2 \begin{cases} < -\sigma, & for\ CBL \\ > +\sigma, & for\ SBL. \\ else, & for\ NBL \end{cases} \quad (1)$$

Here $\sigma$ is the stability threshold of the near-surface potential temperature stratification. Since the neutral stratification
condition ($\sigma = 0$) is rare in nature, consistent with Liu and Liang (2010), $\sigma$ is set to 1.0 K. The threshold value of the NBL is
set to -1.0 to 1.0. Consequently, SBLs and CBLs with weak stable or unstable stratification are possibly identified as NBLs.

Once the boundary layer regime has been identified, we use the criteria defined by Liu and Liang (2010) to estimate the
ABLH for each regime. Since buoyancy is the dominant mechanism driving turbulence in the CBL, the ABLH is defined as
the height at which an air parcel rising adiabatically from the surface becomes neutrally buoyant (Stull 1988). First, we find
the lowest level ($k_1$) (Fig. 2a) that meets the following condition
$$\theta_{k_1} - \theta_1 \geq \sigma_u, \quad (2)$$

in which $\sigma_u$ is the $\theta$ increment that represents the minimum strength of the unstable layer. Once level $k_1$ is determined,
another upward scan is performed to find the lowest level at which the potential temperature gradient with height ($\dot{\theta}_k$) meets
the following criteria
$$\dot{\theta}_k \equiv \frac{\partial \theta_k}{\partial z} \geq \dot{\theta}_r. \qquad (3)$$
Here $\dot{\theta}_r$ is the minimum strength for the overlying inversion layer and can be considered as the overshooting threshold of
the rising parcel to define the scope of the entrainment zone for the CBL. The same procedure is adopted to determine the
NBL height excluding the entrainment zone at the top (Fig. 2a). Various values of $\sigma_u$ and $\dot{\theta}_r$ will affect the determination of
the boundary layer height and they are respectively set to 0.5 K and 4.0 K km$^{-1}$, consistent with Liu and Liang (2010).
Quantifying the uncertainty of the rawinsonde-based approach for identifying ABLH is important, which is closely related to
the thermodynamic characteristics of the sounding profiles (Seidel et al., 2010, 2012; Davy, 2018; Lee and Pal, 2021). The
ABLH determined by this potential temperature gradient method from soundings is highly consistency with that derived
from lidar measurements, with a correlation coefficient of 0.96 and root-mean-square error of 211 m (Liu and Liang 2010).
Moreover, the changes in ABLHs are ≤ 177 m when using 3.5, 4, and 4.5 K km$^{-1}$ as $\dot{\theta}_r$, respectively (Zhang et al., 2017). It
is evident that the uncertainties of the above procedure can be negligible for both CBL and NBL, since most of their ABLHs
are much higher. For the SBL, the turbulence in the ABL can result from either buoyancy forcing or wind shear. The SBL
height is defined as the lower of the heights of both the thermal stable layer from the surface and the maximum wind in the
low-level jet stream if present. More details of the definitions of the boundary layer regimes may be seen in Liu and Liang
(2010). Figure 2c-d shows the typical profiles of potential temperature for CBL, NBL, and SBL at 20:00 BJT on June 10,
2013, July 21, 2013, and August 11, 2013 at Lasa station, and the ABL heights calculated by the potential temperature
gradient method are 3465, 1258, and 409 m AGL, respectively.

## 178 3 Characteristics of the summer ABLH in the eastern and western TP

### 179 3.1 A general characteristic of the ABLH

The diurnal variation is an important feature of the ABL, consisting of different periods of daytime, night−time, and
day/night transitions (Liu and Liang, 2010). In the central TP (near 90 °E), 08:00 BJT, 14:00 BJT, and 20:00 BJT correspond
to 06:00 (the early morning), 12:00 (noon), and 18:00 (the late afternoon) local solar time (LST) (Fig. 1b-c), respectively. To
reveal a difference in ABLH between the eastern TP (ETP) and the western TP (WTP), we divide all sounding stations in the
TP into two groups. One is for the WTP (to the west of 92.5 °E) with 8 stations, and the other is for the ETP (to the east of
this longitude) with 11 stations.

Figure 3a-c shows the spatial distributions of the mean ABLH over the TP at 08:00 BJT, 14:00 BJT, and 20:00 BJT,
respectively. In the early morning (08:00 BJT), the ABL is of the night-time property. The ABLH is generally low (<450 m
AGL) over the TP and displays a relatively homogeneous feature (Fig. 3a). At this moment, the distribution of the ABLH is
narrow, with a frequency peak of 35% at the ABLH of 300 m AGL (Fig. 3d) and 78.5% (99.6%) of the ABLH below 500
(1000) m AGL (Fig. 3e). Figure 3f displays the zonal sections of the ABLH along 32 °N, in which the cross section includes
SQH, GZ, SZ, NQ, CD, GanZ, and HY stations. In this figure, the ABLH varies between 218.4 and 433.9 m AGL from east
to west and presents a relatively homogeneous feature in the west-east direction.

At noon (14:00 BJT), with the well-developed daytime ABL (Fig. 3b), its height remarkably increases over the TP with an
average of 1887.7 m AGL and exhibits a large west-east difference. There is a wide distribution of the ABLH up to 4000 m
AGL, with a relatively flat peak between 900 and 2900 m AGL (Fig. 3d) and only 17.8% (more than 50%) of the ABLH
below 1000 (above 1900) m AGL (Fig. 3e). At this moment, the regional mean ABLH is 2124.2 m AGL in the WTP and
1693.5 m AGL in the ETP, with a mean difference of 430.7 m between the WTP and the ETP. Along 32 °N, the ABLH
remarkably increases from 1379.4 m AGL at GanZ station to 2504.2 m AGL at SQH station, with the west-east difference
exceeding 1200 m (Fig. 3f). This regional difference in the TP ABLH could be likely related to the hydrologic factors such
as air moisture and soil water (also see Section 4) that may modulate the spatial distribution of the daytime ABLH (Seidel et
al., 2012).

In the late afternoon (20:00 BJT), the ABL begins to show the night-time features. The ABLH also starts to decrease in the
ETP, with the regional mean height < 1000 m AGL, while it continues to increase at the west-most stations, with the regional
mean height > 2000 m AGL (Fig. 3c). This result indicates a larger west-east difference (1054.2 m) between the WTP and
the ETP. Especially, the ABLH is 602 m AGL at HY station and 2920.6 m AGL at SQH station, with a difference above
2000 m between these two stations (Fig. 3f). At this moment, the frequency of the high ABLH decreases, with 12.8% of the
frequency peak at the ABLH of 300 m AGL (Fig. 3d) and 50% of the ABL heights less than 1000 m AGL (Fig. 3e). It is
evident that the west-east difference of the ABLH over the TP increases from noon to the late afternoon. During the evening
transition, the daytime boundary layer undergoes a transition to the night-time boundary layer. Since the TP spans almost 1.5
time zones from west to east (Fig. 1c), the local solar time is earlier in the west (where 20:00 BJT corresponds to 17:20 LST
in the westernmost SQH station) compared to the east (where 20:00 BJT corresponds to 18:50 LST for the easternmost HY
station), which supports an earlier transition from the daytime ABL to the night-time ABL in the east (Seidel et al., 2010,
2012; Guo et al., 2016; Lee and Pal, 2017). Meanwhile, it is noted that this difference in the local time is less over TP than
over China (Guo et al., 2016) and the United States (Seidel et al., 2010, 2012; Lee and Pal, 2017). Thus the contribution of
the time zone difference to the regional difference of ABLH is relatively smaller in TP.

Figure 4 further shows the variations of the ABLH from 08:00 BJT to 14:00 BJT and from 14:00 BJT to 20:00 BJT,
indicating varying rates in 6 h. It is seen from Fig. 4a that the ABLH in the TP increases substantially from 08:00 to 14:00
BJT, with a mean growth rate of 1500 m/6 h. There is also a large west-east difference of the ABLH growth rate in this
period, with the regional mean of 1800 m/6 h and 1300 m/6 h in the WTP and the ETP, respectively. From 14:00 to 20:00
BJT (Fig. 4b), the growth rate of the ABLH is negative in the ETP, exhibiting an opposite trend to that in Fig. 4a, which
indicates a significant decrease (around -600 m/6 h) of the ABLH after noon. In the WTP, the growth rate generally shows a
weak increase (around 400 m /6 h) or decrease (around -140 m /6 h). It is evident that the growth rate from 08:00 to 14:00
BJT may indicate the amplitude of the ABL diurnal variation over the TP. Compared to the ETP, the ABL in the WTP has
the larger amplitude of the diurnal variation and the longer development time.

## 3.2 Characteristics of SBL, NBL, and CBL heights

We further examine the characteristics of different ABL types. Figure 5 presents the spatial distribution of occurrence
frequency of SBL, NBL and CBL at 08:00 BJT, 14:00 BJT, and 20:00 BJT. It is seen that the occurrence frequency exhibits
significant discrepancies at different times for the SBL and CBL. At 08:00 BJT, the occurrence frequency of the SBL/CBL is
large/little (Fig. 5a/Fig. 5g), with a mean value 84.9%/8.5% over the TP. At 14:00 BJT, the occurrence of the SBL/CBL
remarkably decreases/increases, accounting for 3.1%/76.9% of the ABL (Fig. 5b/Fig. 5h). At 20:00 BJT, the SBL/CBL
mainly occurs in the ETP/WTP (Fig. 5c/Fig. 5i), with a regional mean of 35.0%/65.0%. However, the NBL shows a
relatively weaker temporal variation over the TP (Fig. 5d-f), with the mean occurrence frequency of 6.4%, 20.0%, and 25.5%
at 08:00 BJT, 14:00 BJT, and 20:00 BJT, respectively. The above results are consistent with the diurnal development of the
ABL structure including the SBL in the early morning, the CBL at noon, and different types of ABLs between the eastern
and western TP in the late afternoon because of the latitudinal difference and the resultant difference in local solar times.
Note that the observations were made simultaneously for all stations. Nevertheless, the daytime SBL and the night-time CBL
may also occur with low frequencies in the TP, which is likely due to the 'abnormal' forcing associated with certain synoptic
conditions or cloud coverage (Medeiros et al., 2005; Poulos et al., 2002; Stull, 1988).

To analyse the temporal variations of the ABLH over the TP, the ABL height-occurrence frequency relationships for the
SBL, NBL, and CBL at 08:00 BJT, 14:00 BJT, and 20:00 BJT are presented in Fig. 6a-f. For the SBL, the frequency
distribution of the ABLH shows the similar feature at three measurement times (Fig. 6a-c) and is characterized by a narrow
single mode, with the frequency peaks of 39.0%, 28.1%, and 36.6% at the ABLH of 200, 300, and 300 m AGL at 08:00,
14:00, and 20:00 BJT, respectively, which indicates small temporal variations of the SBL height due to its turbulence
inhabited. Moreover, the SBL height above 80% is < 600 m AGL and the cumulative frequency of the SBL height exceeding
1000 m AGL is little (near zero) at 08:00, 14:00, and 20:00 BJT (Fig. 6d, e, and f). For the NBL and CBL, however, their
heights vary strongly with time under the influence of surface heating in the daytime. At 08:00 BJT (Fig. 6a), the
distributions of the NBL and CBL heights are narrow, with the frequency peaks of 27.5% and 35.1% at the ABLH of 300 m
AGL for NBL and CBL, respectively, similar to that of the SBL, which is possibly due to the initial development of the CBL
and NBL in the early morning. At 14:00 BJT, the CBL and NBL have a wide distribution of the ABLH up to 4000 m AGL,
with a relatively flat peak between 1000 and 3000 m AGL, which is remarkably different from a single peak of the SBL. The
frequency of the NBL height between 500 and 3000 m AGL is generally less than 5% (Fig. 6b), with a frequency peak of 6.1%
at 1000 m AGL, and more than 50% NBL height exceeds 1700 m AGL (Fig. 6e). The height of the CBL is higher, with a
frequency peak near 4.5% between 1500 and 2500 m AGL (Fig. 6b) and more than 50% CBL height is above 2000 m AGL
(Fig. 6e). These results show that the ABL develops well at noon. When the ABL begins turning to the nigh-time property at
20:00 BJT (Fig. 6c and 6f), the distributions of the CBL and NBL heights are still wide but the frequency of the high ABL
height decreases, with the frequency peak below 500 m AGL. It is obvious that the CBL and NBL heights show the similar
results consistent with those from Zhang et al (2017). Stull (1988) and Blay-Carreras et al. (2014) revealed that the NBL
often occurs in the transition periods between the CBL and the SBL. Since these transitions occur rapidly, the NBL may
have the same characteristics in the state variables as the CBL prior to the transition although the dynamic forcing in the
NBL maybe weaker compared to the CBL.

To reveal the spatial variations of the ABLH over the TP, the distributions of mean SBL, NBL, and CBL heights at 08:00
BJT, 14:00 BJT, and 20:00 BJT are illustrated in Fig. 7. The SBL height is generally low and varies between 200 and 730 m
AGL at these times, with a mean height of 336.0 m AGL at 08:00 BJT, 356.0 m AGL at 14:00 BJT, and 321.9 m AGL at
20:00 BJT (Fig. 7a-c), which indicates the weak spatial differences of the SBL height over the TP at three observation times.
For the NBL and CBL, their heights are still low in the early morning (Fig. 7d and 7g), with the ABLH < 450 m AGL, and
have small spatial differences. At noon (Fig. 7e and 7h), the NBL and CBL heights rapidly increase, especially in the WTP,
which leads to a remarkable east-west gradient in the ABL height. At this moment, there is a regional mean NBL/CBL
height of 2074.6/2191.4 m AGL in the WTP and 1594.8/1788.0 m AGL in the ETP, with a difference of 479.8/403.4 m
between the WTP and the ETP. In the late afternoon (Fig. 7f and 7i), the NBL/CBL height continues to increase in the WTP,
with a regional mean of 2092.0/2192.2 m AGL, while the NBL/CBL height begins decreasing in the ETP, with a regional
mean of 1423.1/1237.2 m AGL. This varying feature in the ETP and WTP results in the larger differences of 668.9/955.0 m
in the NBL/CBL height between the WTP and ETP. Thus there is a significant difference in the frequency distribution of the
ABL height between the ETP and the WTP in the daytime (Fig. 6g). The cumulative frequency contours gradually go
upward from east to west (Fig. 6h). The eastern TP is dominated by a low CBL height, with the peak of 14.4% at the height
of 350 m AGL (Fig. 6g) and the 50% (5%) CBL height below 1000 m AGL (above 2500 m AGL) (Fig. 6h). For the WTP,
the strong peak of 4%-10% corresponds to the high CBL between 2500 and 3500 m AGL (Fig. 6g), especially at SQH
station, and there are larger CBL heights, with almost 50% CBL extending upward to more than 2500 m AGL, almost 10%
reaching 4000 m AGL or higher, and only 15% CBL below 1000 m AGL (Fig. 6h).

To investigate an effect of differences in the sample profiles shown in Fig. 1b and d, we use the test group dataset to repeat
the above analyses. Figures 8a and b show the scatter plots of the occurrence frequency of the SBL, NBL, and CBL from the
original and test group datasets at each of 19 stations at 08:00 BJT and 20:00 BJT, respectively. It is seen that the correlation
coefficients between the two datasets are 0.92-0.99, with root-mean-square errors (RMSEs) of the occurrence frequency
between 1.1% and 2.7%. The similar results are also seen in the SBL, NBL, and CBL heights at 08:00 BJT (Fig. 8c) and
20:00 BJT (Fig. 8d). The correlation coefficients in the ABL height are 0.90-0.99. The RMSE of the SBL height is 14 m and
25 m at 08:00 and 20:00 BJT, respectively. The RMSE of the CBL and NBL heights are 54-59 m at 08:00 BJT and 99-107m
at 20:00 BJT. These high correlations and small errors show that the difference in the sample size does not change our
conclusions.

From the foregoing analysis, the CBL and NBL heights in the TP show remarkable temporal variations and west-east spatial
differences, while these features are not remarkable for the SBL. From noon to the late afternoon, the NBL and CBL are
deeper in the WTP compared to the ETP, with the ABLH difference between the WTP and the ETP exceeding 600 m AGL
at 20:00 BJT. Then, which factors contribute to this difference in the ABL between the WTP and ETP? In the following
section, we examine some factors that may be responsible for the ABL height over the TP.
**4 Factors responsible for the ABL height over the TP**
Previous studies have addressed effects of surface sensible heat flux (SHF), soil volume moisture content (VWC), downward
solar radiation flux (DSR), and the cloud cover (CLD) on ABL height (Liu, et al., 2004; Zhao et al., 2011; Sanchez-Mejia
and Papuga, 2014; Rihani et al., 2015; Lin et al., 2016; Zhang et al., 2017; Zhang et al., 2019; Qiao et al., 2019). However,
these studies paid little attention to reasons for the west-east difference of the ABL between the eastern and western TP. To
investigate a possible reason for this difference, we utilize the TIPEX-III SHF, DSR, and VWC at SQH, NQ, and LZ stations,
and the corresponding meteorological operational CLD observations to analyze the relationships between these variables and
the ABL height.

The driving force of turbulence in the ABL is the surface buoyancy flux as a result of surface and air temperature and
humidity differences and the mean surface layer wind. The kinematic heat flux (KHF) and kinematic moisture flux (KMF) at
the surface are the two directly factors responsible for the surface buoyancy flux (Brooks and Rogers, 2006). Since KMF is
often small, KHF associated with SHF is examined as a major component of buoyancy flux in dry conditions over land.
According to the method of Brooks and Rogers (2006), our calculation results show that the contribution from KMF to
surface buoyancy flux is below 18% at SQH, NQ, and LZ stations. Moreover, the ABL may be largely affected by the effect
of cumulative SHF in the daytime (Zhang et al., 2019). Thus we analyse the possible effect of SHF on the ABL. Figure 9a-c
presents the scatter plots between the mean SHF over the past six hours and the ABL height at SQH, NQ, and LZ stations.
As shown in this figure, the correlation is 0.80, 081, and 0.71 (significant at the 99% confidence level) at these stations,
respectively. When SHF is strong, the turbulent motion is strong and the ABL develops, which is consistent with the result
of Zhang et al. (2011b). Their result shows a significant correlation of 0.78 in the arid area of Northwest China between the
ABL thickness and the cumulative SHF. Figures 10a and b further present the features of the ABL height and SHF at SQH,
NQ, and LZ stations. The mean value of SHF is 85 W/m$^2$, 42 W/m$^2$, and 33 W/m$^2$ at SQH, NQ and GZ stations, respectively,
and has a large difference (52 W/m$^2$) between SQH and NQ stations. This result indicates a decreasing trend of SHF from
west to east in the TP, consistent with a reduction of the ABL height from SQH via NQ to LZ station (shown in Figs. 3 and
10a). In addition, Fig. 11 demonstrates the diurnal variations of SHF and the ABL height at SQH, NQ, and LZ stations. The
duration of positive SHF in a diurnal cycle at SQH, NQ and GZ stations is 14, 12 and 11 hours, respectively, and indicates a
decreasing trend from west to east in the TP. It is clear that the peak of the SHF occurs earlier than the maximum ABLH in a
diurnal cycle at SQH station. The maximum ABL height occurs near 20:00 BJT (approximately 17:20 LST), corresponding
to a strong SHF. At LZ station, however, the SHF turns into a negative value at 20:00 BJT (18:20 LST) and then the ABL
height decreases. Some past studies show that the development of ABL height generally lags the development of SHF, and
ABL depth growth continues even after SHF attains the maximum daytime value until the time of early evening transitions
(Chen et al., 2016; Zhang et al., 2019). Consequently, the difference in the ABL height between the WTP and ETP is closely
associated with a west-east difference in SHF that is as a direct thermal factor for the ABL development in the TP.

The solar radiation at the surface is an important component of the surface energy budget, affecting surface temperature and
SHF. We show the scatter plots between the 6-hour mean DSR and the ABL height at SQH, NQ, and LZ stations (Fig. 9d-f).
The ABL height is highly correlated with the 6-hour average of DSR at these stations, with the correlation coefficients of
0.86, 081, and 0.73, respectively, which is equivalent to those of SHF. The mean DSR shows a decreasing trend from SQH
(510 W/m$^2$) to LZ (200 W/m$^2$) station. Since the solar irradiance at the surface is negatively associated with the local cloud
cover (Guo et al., 2016; Lin et al., 2016; Li et al., 2017; Zhang et al., 2017), the cloud cover is also correlated to the ABL
height. Figure 9g-i shows that the 6-hour mean CLD has significant correlations of -0.56, -0.65, and -0.54 with the ABL
height at SQH, NQ, and LZ stations, respectively. A decrease of the mean ABL height from SQH to LZ station (Fig. 10a) is
corresponded to an increase of cloud cover (Fig. 10d) and a decrease of DSR (Fig. 10c). When cloud cover is between 0 and
20%, the mean ABL height for the NBL and CBL is 2019 m AGL/2732 m AGL in the ETP/WTP; and when cloud cover
is >80%, the ABL height decreases to 741 m AGL/1626 m AGL in the ETP/WTP (Fig. 12). Therefore, the increased cloud
cover inhibits the development of both the NBL and CBL. The difference in cloud cover between the WTP and ETP
contributes to the west-east distribution of DSR and SHF, also finally contributing to the difference of the ABL development.
Corresponding to more cloud cover in the ETP, the local ABL is more closely associated with atmospheric moisture
processes.

Soil moisture is also an important factor affecting SHF. Low soil moisture generally coincides with a high surface sensible
heat flux, which facilitates the ABL development (e.g., McCumber and Pielke, 1981; Sanchez-Mejia and Papuga, 2014;
Rihani et al., 2015). Figure 9j-l shows that the relationship between the ABL height and the 6-hour mean VWC at SQH, NQ,
and LZ stations. The ABL height at LZ station is negatively correlated to the local soil moisture, with a significant
correlation coefficient of -0.45. This result indicates that the ABL height is lower when surface soil is moister. However, the
negative correlation is weaker at SQH station, with a correlation coefficient of -0.21. This difference between the WTP and
the ETP may be associated with the climatic feature of the local soil moisture. The surface type transitions from alpine
meadow with few shrubs and trees or alpine steppe in the ETP to bare soil with few obstacles in the WTP (Wang et al.,
2016). Accordingly, soil moisture decreases gradually from the ETP to the WTP (Fig. 10e), with a mean value of soil
moisture below 0.10 $m^3/m^3$ at SQH station and 0.38 $m^3/m^3$ at LZ station. Little soil moisture in the WTP has a weak
modulation to the local surface heat flux, which may lead to a weak correlation between the ABL height and soil moisture in
the WTP.
**5 Summary and discussion**
Using the summer TIPEX-III intensive and meteorological operational observational datasets, we examine the ABL features
and the relationships of the ABL height with surface sensible heat flux, solar radiation, cloud cover, and soil moisture in the
TP region. The main conclusions are summarized as follows.

Generally speaking, the ABL height exhibits diurnal variations and regional differences in the TP, especially for the CBL
and NBL. These features are weak for the SBL. Compared to the ETP, the ABL in the WTP has the larger amplitude of the
diurnal variation and the longer development time. In the early morning, the ABL height is generally low over the TP, not
showing a large west-east difference, and the distribution of the ABL height is narrow, with 78.5% of the ABL height < 500
m AGL. At noon, the CBL and NBL heights remarkably increase and have a wide distribution in the ABL height up to 4000
m AGL, with more than 50% of the ABL height exceeding 1900 m AGL. Their heights exhibit a large west-east difference.
At this moment, the distribution of the SBL height is also narrow, with the cumulative frequency of 80% at the height of 500
m AGL, and there is no remarkable west-east difference. In the late afternoon, there are a narrow distribution of the SBL
height and wide distributions of both the NBL and CBL heights. At this moment, the ABL height continues to increase in the
WTP, while it begins to decrease in the ETP. This feature results in a larger west-east difference in the ABL height. In spite
of a cold environment in the TP (relative to plain areas), the WTP still has the ABL height above 2000 m AGL, which is
similar to some extreme hot and arid areas such as Dunhuang and Taklimakan Deserts. In the ETP, the ABLH is similar to
that in North China (1500-1900 m AGL) and is generally larger compared to the East Asian summer monsoon region (<
1500 m AGL) such as the Yangtze River Delta and the Pearl River Delta (Zhang et al., 2011; Guo et al., 2016; Zhang et al.,
2017; Qiao et al., 2019).

The occurrence frequency of the SBL and CBL in the TP shows remarkable temporal variations. Most (few) of the SBLs
(CBLs) occur in the early morning and the occurrence frequency rapidly decreases (increases) at noon, accounting for 3.6%
(76.9%) of the ABL in the TP. Possibly owing to a difference in the solar elevation angle with respect to longitude in the late
afternoon, the SBL and CBL dominate the ETP and WTP, respectively. However, the NBL shows a relatively weak temporal
variation over the TP, with the mean occurrence frequency of 6.4% in the early morning and around 20% at noon and in the
late afternoon.

The ABL height is significantly correlated to SHF, DSR, and cloud cover in the TP and is also closely associated with soil
moisture in the ETP. The decreasing trends in both SHF and DSR and the increasing trends in both cloud cover and soil
moisture from west to east may cause the corresponding west-east reduction in the ABL height. In the WTP (ETP), with low
(high) cloud cover, there is larger (smaller) downward solar radiation at the surface. Meanwhile, corresponding to bare soil
(alpine meadow or steppe) in the WTP (ETP), there is a dry (wet) soil condition. These features cause high (low) sensible
heat flux, thus promoting (inhibiting) the local ABL development. The above factors affecting the WET and ETP ABL
heights are summarized in Fig. 13.

The Tibetan Plateau is an area very sensitive to global climate change, which exerts important thermal and dynamical effects
on the general circulation and climate through the unique and complex land surface and boundary layer processes. Owing to
new sounding observations in the WTP, our analysis firstly reveals remarkable west-east differences in the ABL height,
occurrence frequency, and diurnal amplitude over the TP region during summer. It is noted that there is a big drop in the
CBL height from 3000–4000 m AGL to 1000–2000 m AGL from the WTP to the ETP. Such a steep west-east
inhomogeneity in the TP (with an East-West spatial scale of about 2000 km) is remarkably different from the regional
variability of the ABLH on much larger scales (~4000 km) such as in the United States (Seidel et al., 2012) and in China
(Guo et al., 2016). This unique inhomogeneity in the TP may trigger the local mesoscale circulation and precipitation (Segal
et al., 1992; Goutorbe et al., 1997; Huang et al., 2009; Zhang et al., 2019; Qiao et al., 2019). Therefore, the influences of the
west-east differences in the ABLH over the TP on the local weather and climate should be further studied in the future.  In
addition, this study merely investigates the characteristics of the summer ABLH in TP due to the limitation of the intensive
sounding observations. More efforts should be made to expand the climatology of ABLH to other seasons in TP when more
sounding data are available.

Code and data availability. All data used are available from the TIPEX-III on its homepages (http://data.cma.cn/tipex).

Author contributions. J.C. designed the study, analyzed the data and wrote the manuscript. P.Z. contributed to the study
design, supervisor, and writing of the manuscript.

Competing interests. The authors declare that they have no conflict of interest.

Acknowledgements. We thank the TIPEX-III for providing the data available on its homepages (http://data.cma.cn/tipex).
This work is supported by the National Key Research and Development Program of China and the Strategic Priority
Research Program of Chinese Academy of Sciences.

Financial support. This work is jointly funded by the National Key Research and Development Program of China (Grant
2018YFC1505700) and the Strategic Priority Research Program of Chinese Academy of Sciences (XDA20100300).

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

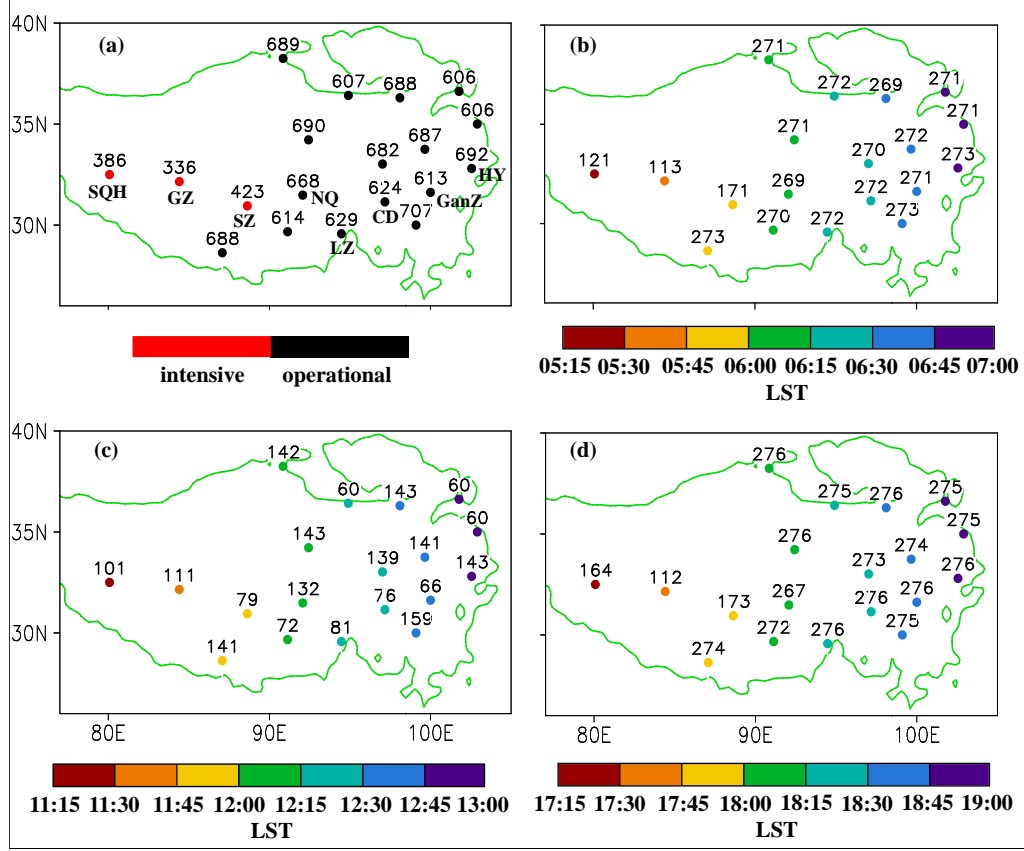


**Figure 1: Distribution of sounding stations, in which the number indicates sounding profiles at each station at (a) 08:00, 14:00, and 20:00 BJT, (b) 08:00 BJT, (c) 14:00 BJT, and (d) 20:00 BJT in the study period. Red (black) dots represent intensive (operational) observations, and some observation station names are given as abbreviations in (a). Colored dots represent the local time of the BJT time in (b), (c) and (d). The green line shows the 3000 m topography.**



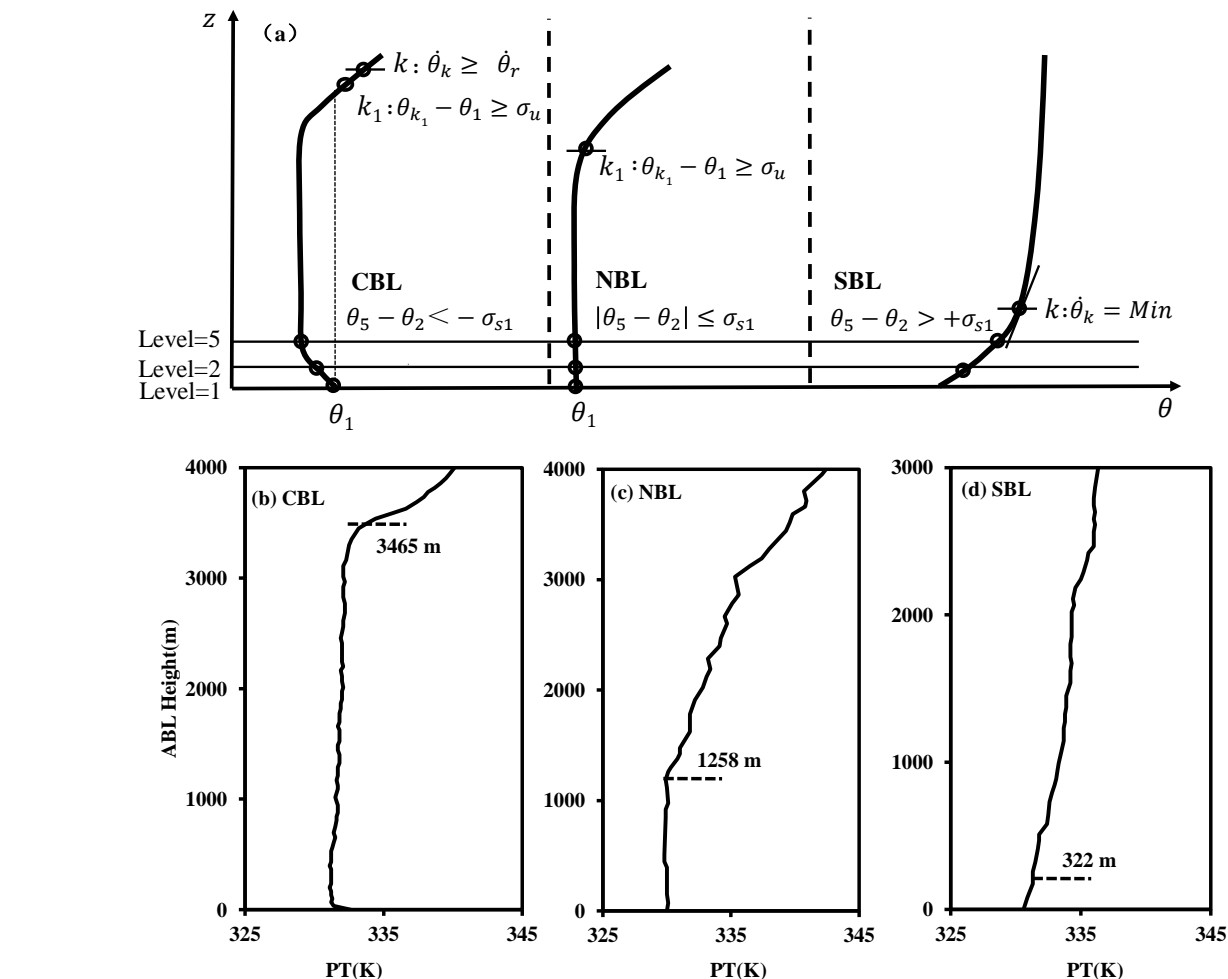



Figure 2: (a) Illustration of the determination procedure for the convective boundary layer (CBL), neutral boundary layer (NBL), and stable boundary layer (SBL) heights; and examples of the potential temperature (PT) profiles derived from sounding observation at Lasa station at 20:00 BJT for (b) CBL on June 10, 2013, (c) NBL on July 21, 2013, and (d) SBL on August 11, 2013, respectively. The dash line in (b)-(d) represents the ABL height identified using the algorithm described.


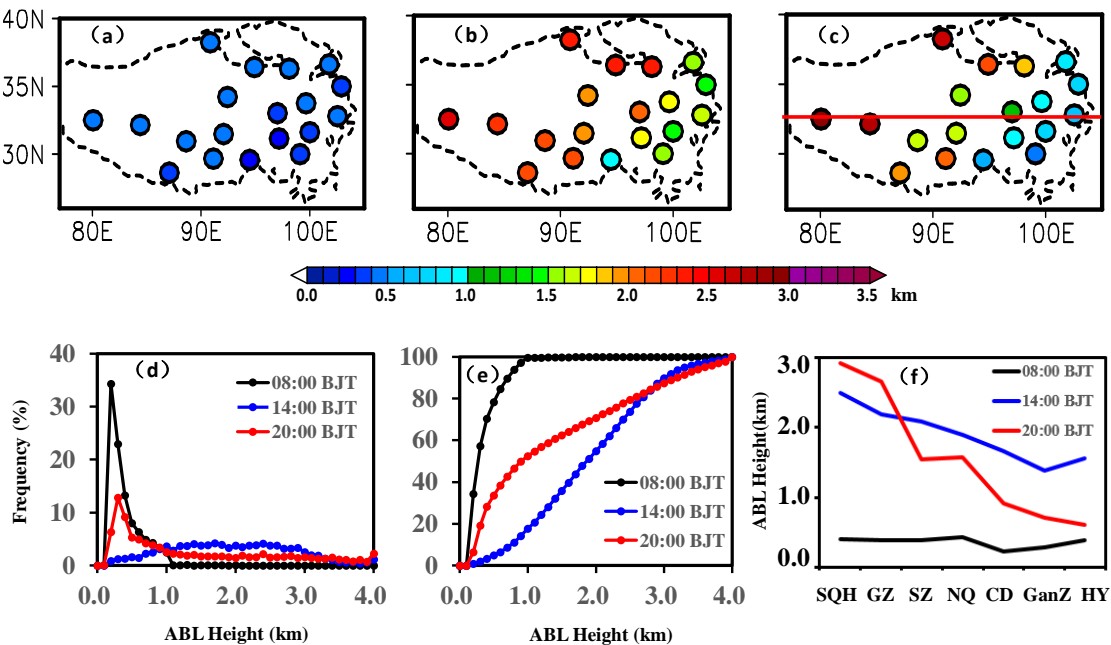


**Figure 3:** Spatial distribution of the mean ABL height (ABLH) at (a) 00:08 BJT, (b) 14:00 BJT, and (c) 20:00 BJT; (d) the regional mean frequency and (e) cumulative frequency distributions of the ABLH in the TP at 08:00 BJT, 14:00 BJT, and 20:00 BJT; (f) the west-east cross sections of the ABLH along 32°N (indicated by red line in (c)) at 08:00 BJT, 14:00 BJT, and 20:00 BJT.


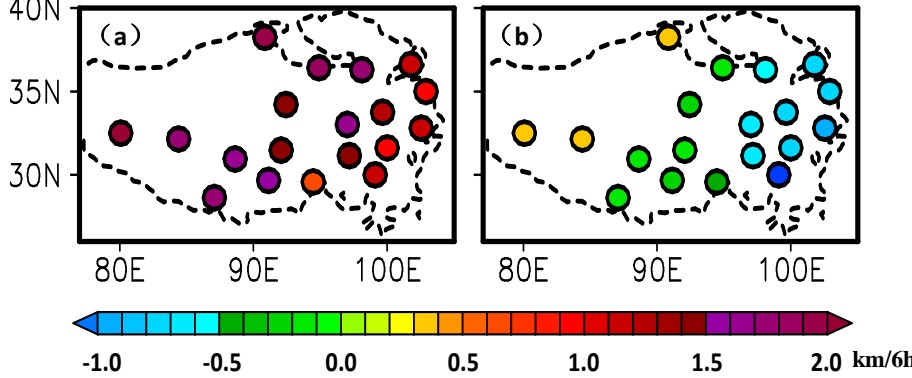


**Figure 4:** Spatial distribution of the ABLH growth rate from 08:00 BJT to 14:00 BJT (a) and from 14:00 BJT to 20:00 BJT (b).


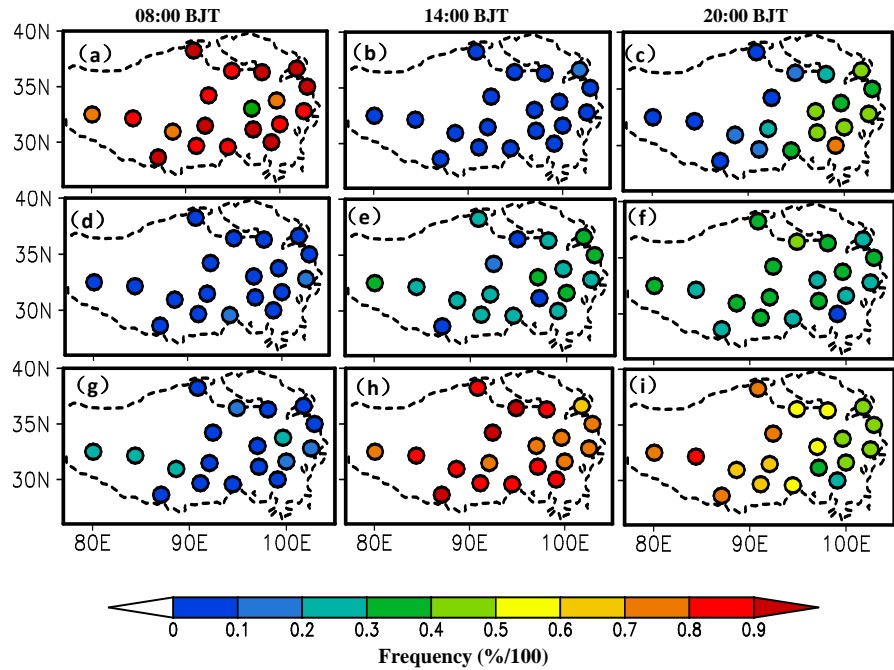

**Figure 5: Spatial distribution of the occurrence frequency for the SBL (top), NBL (middle), and CBL (bottom) at 08:00 BJT, 14:00 BJT, and 20:00 BJT.**

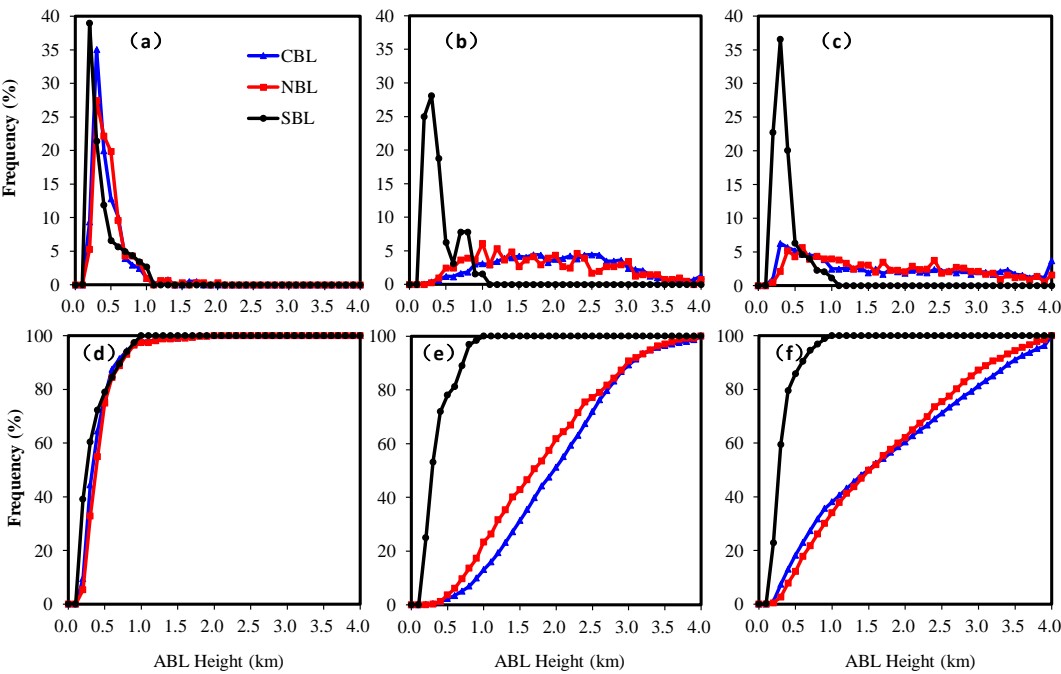

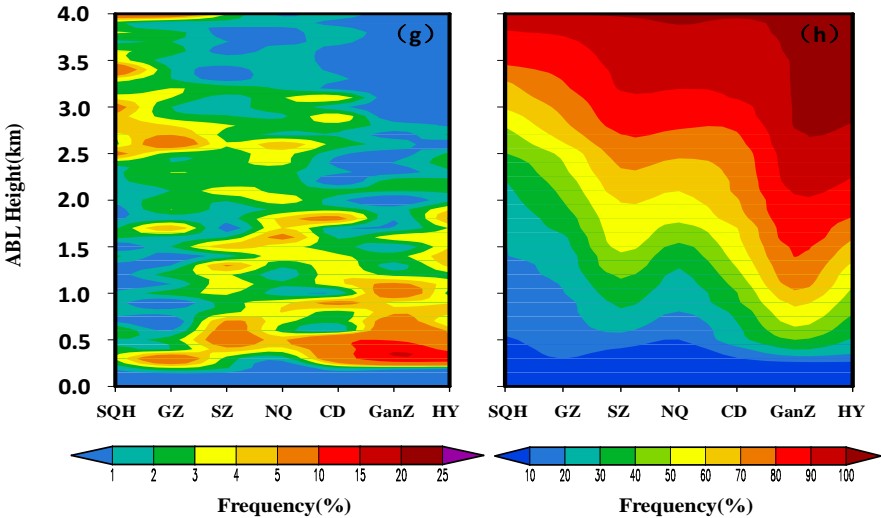

616

**Figure 6: The regional mean frequency distributions of the ABLH over the TP for the CBL (blue), NBL (red), and SBL (black) at (a) 08:00 BJT, (b) 14:00 BJT, and (c) 20:00 BJT; and (d)-(f) same as in (a)-(c) but for the cumulative frequency distributions; and the west-east cross sections of frequency (g) and cumulative frequency (f) distributions of the CBL height along 32°N in the daytime (14:00 and 20:00 BJT).**

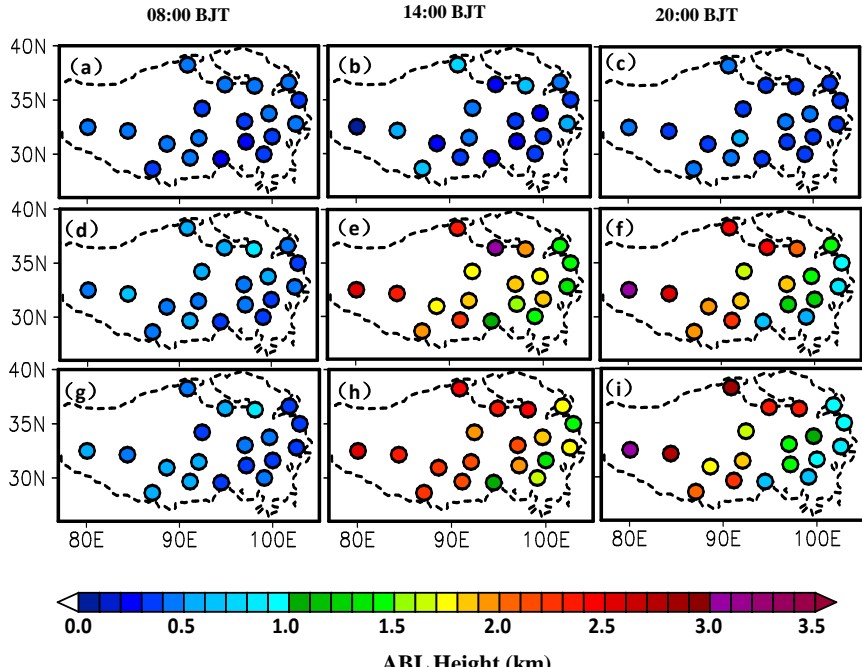



**Figure 7: Spatial distributions of the mean ABLH for the SBL (top), NBL (middle), and CBL (bottom) at 08:00 BJT, 14:00 BJT, and 20:00 BJT.**


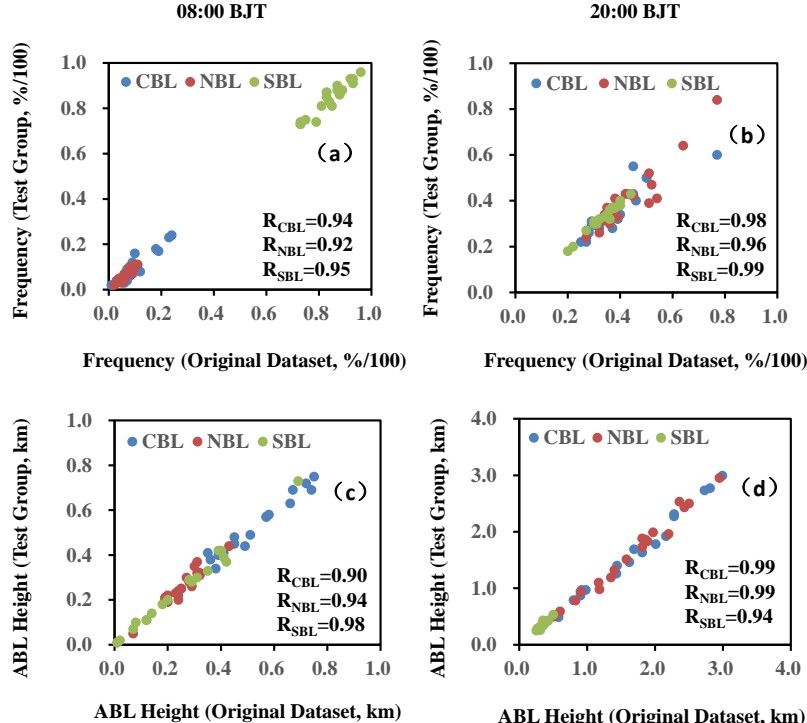


Figure 8: The scatter plots of occurrence frequency of the SBL, NBL, and CBL for the original and test group datasets at 19
stations at (a) 08:00 BJT and (b) 20:00 BJT; and (c)-(d) same as in (a)-(b) but for the ABLH.

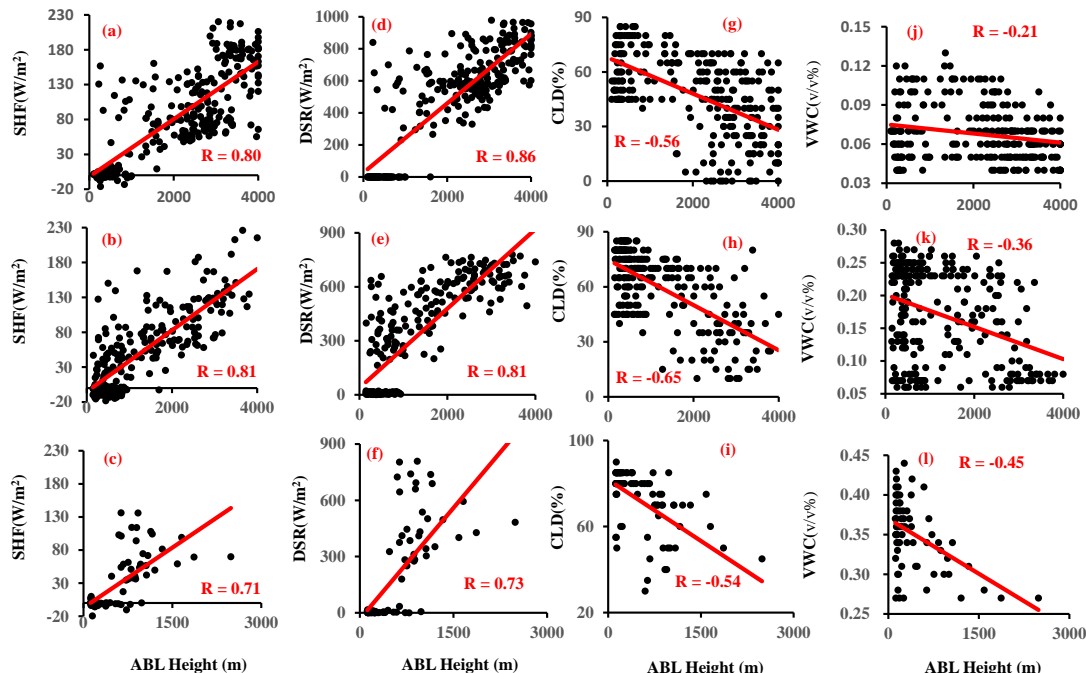


**Figure 9: Scatter plots of the ABLH and the 6-hour average of surface sensible heat flux (SHF) (a-c), surface downward solar irradiance (DSR) (d-f), total cloud coverage (CLD) (g-i), and surface soil volume moisture content (VWC) (j-l) at 08:00 BJT, 14:00 BJT, and 20:00BJT at SQH (top), NQ (middle), and LZ (bottom) stations in the study period. The correlation coefficient (R) is given in each panel.**

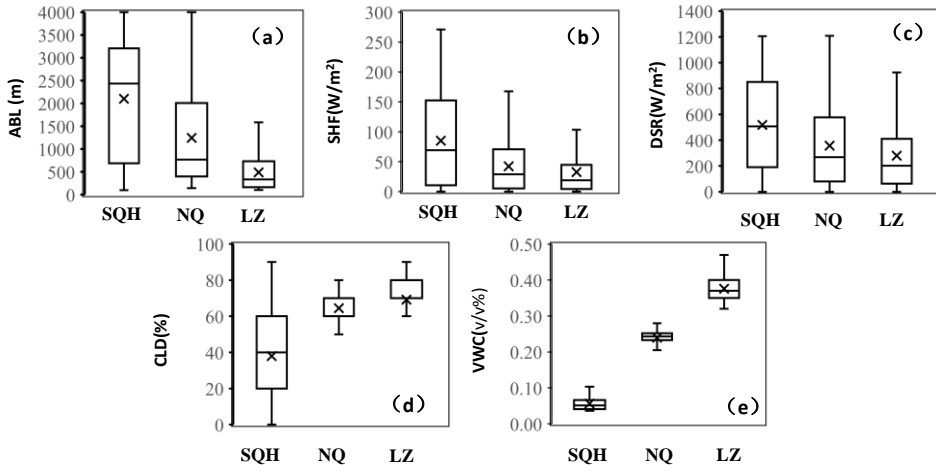


**Figure 10: (a) The ABLH, (b) SHF, (c) DSR, (d) CLD, and (e) VWC at SQH, NQ, and LZ stations in the study period. Horizontal bars show the 5th, 25th, 50th, 75th, and 95th percentile values and "×" symbols show the corresponding mean value.**

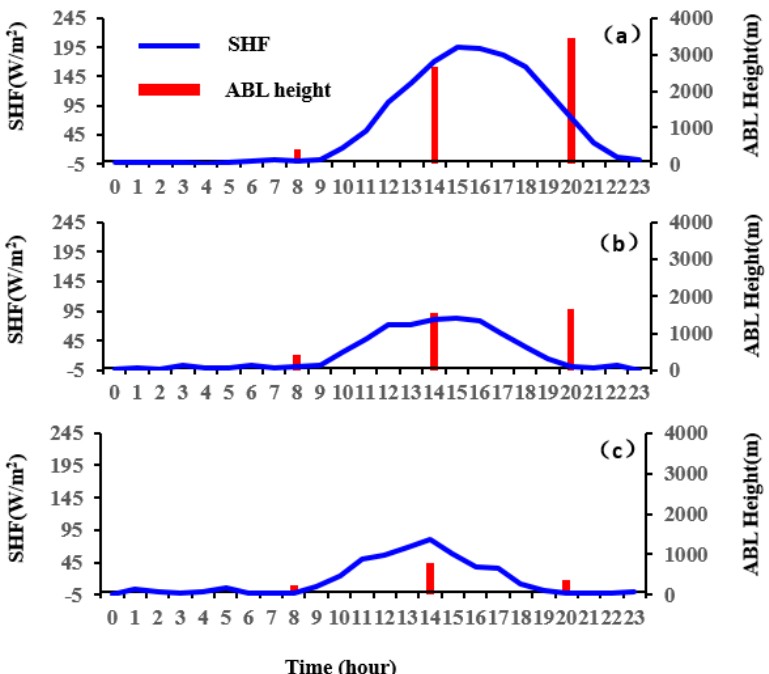


**Figure 11: Diurnal variations of surface sensible heat flux (blue) and the ABLH (red) averaged over the study period at (a) SQH, (b) NQ, and (c) LZ stations.**

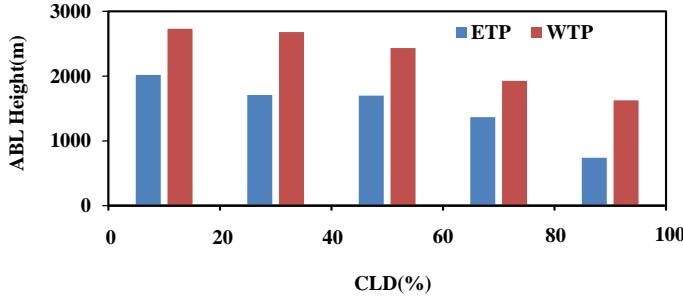


**Figure 12: The mean ABLH (for the NBL and CBL) and CLD over the ETP (blue) and WTP (red) in the daytime (14:00 BJT and 20:00 BJT).**

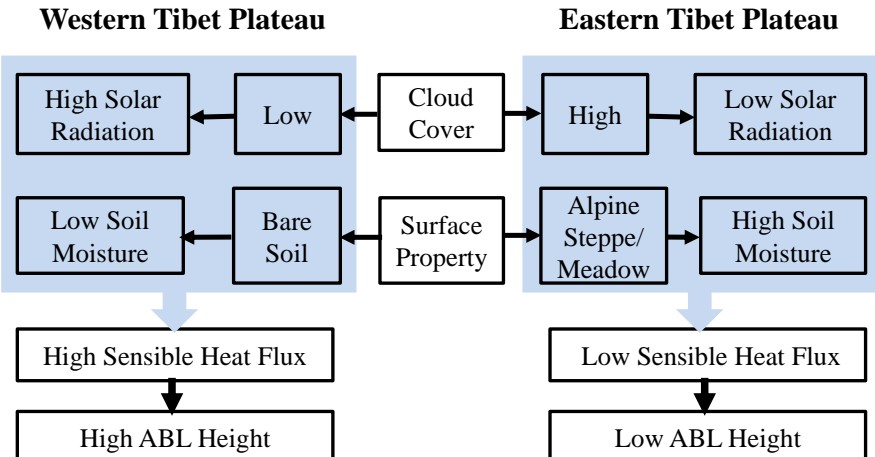


**Figure 13: The schematic diagram for relationships between the ABLH and the influential factors in the ETP and the WTP.**