# Peer review of "Characteristics of the summer atmospheric boundary layer height over the Tibetan Plateau and influential factors"

_Atmospheric Chemistry and Physics, 2020_

## Referee Comment (RC1) · Anonymous Referee #1 · 22 Oct 2020

**General Comments:**

The atmospheric boundary layer (ABL) is the lowest part of the atmosphere in which a variety of complex motions, characterized by turbulence, may be present. In a diurnal cycle, the different manifestations of an ABL are generated by distinct forcing mechanisms that originate from mechanical and thermal effects which impart different depth scales and characteristic velocities to the ABL. For instance, a convective boundary layer (CBL) is defined by its depth zi, which is usually taken as the height of the lowest inversion, by a convective velocity scale w\* and by a friction velocity scale u\*. On the other hand, a stable boundary layer (SBL) is characterized by the depth of the nocturnal

surface inversion (NSI), usually denoted by hi, and u\*. During daytime, the positive turbulent heat flux establishes the structure of the CBL. Conversely, in a night-time stable ABL, the negative turbulent heat flux and clear-air radiative cooling, on average, control the development of the NSI layer. Relevant periods occurring during the existence of an ABL concern transition situations, in which the turbulent heat flux switches sign and assumes positive and negative magnitudes. Over land and after sunrise the surface is warmed and heat is transferred upward by updrafts of warm air. Such a mechanism generates a deepening of the primitive CBL. In late afternoon, during sunset when the external forcing, such as the upward sensible heat flux and geostrophic forcing, varies rapidly, clear-air radiative cooling occurs and the turbulent heat flux decreases and becomes negative. In consequence, the convective turbulence decays with an accelerating rate and starts the formation of the NSI. Above the NSI, the convective energy-containing eddies start to lose their strength and mixing capacity, and the CBL begins to decay. This deep and near-adiabatic vertical region, which is the remnant of the daytime CBL, is known as the residual layer. The use of precise information on this residual layer in numerical models is of fundamental importance when describing the evolution of the diurnal CBL. Generally, experimental campaigns are important when used to describe the physical characteristics of the ABL. These observational studies usually employ automatic meteorological stations and radiosounding to measure the distinct characteristic parameters that allow for an understanding of the turbulent patterns occurring in a diurnal cycle of the ABL. The present article is basically an observational study in which the authors estimate the height of the ABL at different times of the day on the Tibetan Plateau. To carry out this investigation they employ routine operational data and detailed measurements obtained from programmed vertical probes. From the scientific point of view and for practical applications in distinct branches of meteorology this study is important and has meaning. In addition, knowing the height of the ABL allows the environmental impact of polluting sources to be evaluated. However, the study exhibits some deficiencies that need to be corrected so that the results presented and discussed by the authors can be better enjoyed and used by
the readers. Based on what was written above, the authors should better describe the existing types of the atmospheric boundary layer. Thusly, it is necessary and indispensable that the manuscript contains a detailed description of the formation and evolution of the planetary boundary layer. The equation for the potential temperature difference (PTD) on line 115 is very vague and poorly understood. Authors should make a greater effort to characterize the physical criteria that allow choosing and safety to identify the types of the atmospheric boundary layer. It is important to consider that buoyancy effects make the convective and stable ABLs strikingly distinct. Furthermore, the authors should consider in their analysis the fact that "The neutral ABL is rare because small virtual temperature differences in the ABL can cause large buoyancy patterns". How the authors identify this particular type of ABL? The authors also need to build vertical temperature and wind profiles and display them in the study. Associating these vertical profiles with the types of ABL is very important in observational studies. The set of suggestions proposed above will allow readers to accept the observational results with greater reliability.

**Major comments**

Line 115: The PTD classification is a fundamental criterium for the present manuscript. As a consequence, the authors must provide a more detailed discussion of the employed methodology to obtain the heights of the distinct ABL types. As the manuscript is basically observational data analysis, is not enough for the readers the citations presented.

Line 155: How a SBL can occur at noon (14:00 BJT). In this daytime period, there is a CBL. How the CBL height is near to the NBL height? The authors need to clarify.

**Minor comments**

abstract "The SBL accounts for 85% of the TP ABL. At noon, there is a wide distribution in the ABL height up to 4000 m. The CBL accounts for 77% of the TP ABL, with more than 50% of the CBL height above 1900 m." Please rewrite more clearly this statement.
For this reviewer the above statistics are confused.

Line 24: The authors need to present a better definition of the ABL.

Line 154: Please correct the hour "00:80 BJT"

---

## Referee Comment (RC2) · Anonymous Referee #2 · 26 Oct 2020

Review of 'Characteristics of the summer atmospheric boundary layer height over the Tibetan Plateau and influential factors' by Che and Zhao (ACP-2020-787)

This manuscript intends to explore the characteristics of the stable and convective boundary layers in the TP using sounding and surface station measurements from an intensive field campaign and routine sounding stations. The authors used a very unique and valuable dataset and did very thorough analyses on the height, frequency of occurrence, and the physics processes leading to the observed ABL properties. The results revealed the persistent east-west difference in ABL height during the daytime and the day/night transition period. These results have significant implications in revealing the cause of mesoscale circulation and weather systems over the TP. The author further identified different mechanisms for these differences (land surface properties and differences in day/night transition time). They also evaluated the potential impact of sample size on their findings.

Overall, this is a high-quality paper with significant results using a unique dataset. I recommend publication of the manuscript in the ACP after minor revisions suggested below.

Main concerns:

1. Although the presentation of the manuscript has good logic flow, descriptions of the data processing or the results can be confusing at places. The clarity of the manuscript can be improved. I have made some specific suggestions listed in the minor concerns, but the authors should go through the manuscript very carefully or get help from people experienced in writing scientific articles in English.

2. The manuscript can be enhanced if the methodology in defining the ABL types using the PTD are revisited in the discussion section. For the SBL, the mode of the ABL height is around 300 m, suggesting that the PTD represents the temperature gradient in the main body of the stable ABL. For the CBL, since few measurements shows CBL height less than 500 m (except at 20:00 BJT), the 50 m height is likely within the surface layer. The 250 m level, on the other hand, can be in the surface layer or in the well-mixed portion of the CBL depending on the CBL height (assuming the surface layer is ~10% of the ABL). The PTD in this case represent approximately the potential temperature difference in the surface layer or between the surface and the well-mixed CBL. The meaning of the PTD for the NBL should be similar to that in the CBL except with a smaller temperature difference. Clarifications like this should be helpful to the readers.

   Also, how sensitive are the results to the choice of $\sigma$? My general feeling is that their results are not sensitive to the choice of $\sigma$ since the results of the CBL and NBL are very similar. However, the authors should make appropriate comments on the sensitivity issue.

3. The overall results in this manuscript is consistent with the diurnal evolution of the ABL with the daytime deep CBLs and nighttime shallow SBL. There are also occurrences of daytime SBLs and nighttime CBLs although the frequencies of occurrence for both are small. The daytime SBL or nighttime CBL are likely results of 'abnormal' forcing associated with certain synoptic conditions or cloud coverage. The authors mentioned a few times throughout the manuscript about the 'diurnal variations' of the SBL or the CBL

(e.g., Lines 276, 290). These wordings are misleading and should be revised. It would be interesting to look into the mechanisms of the occurrence of daytime SBL and nighttime CBL, but it may be beyond the scope of this paper.

Minor points:

Line 14: 'The SBL accounts for 85% of the TP ABL' should add the time frame here to avoid misunderstanding: 'The SBL observed during this time accounts for 85% of the TP ABL'

Line 15: 'The ABL height exhibits….', again, need to specifiy time: 'The ABL height at noon exhibits…'

Line 20: 'For the western (eastern) TP…' , make it 'In general, for the western (eastern) TP…'

Line 28: change 'convective transmission' to 'convective transport'.

Line 56: change 'have addressed' to 'found'; also change 'can be as high as 2000–3000 m' to 'can reach 2000–3000 m'.

Line 57: change 'Song et al. (1984) examined the ABL height at Gaize station of the western TP is
above 3000 m, while the ABL heights….' To 'Song et al. (1984) found the ABL height at Gaize station of the western TP to be above 3000 m, while the ABL heights….'

Line 62: 'These results show that the ABL height over the TP varies greatly with position and season'. Change 'position' to 'location'.

Line 67: change 'and less-developed logistics' to 'logistic challenges'.
Line 68: remove 'a short-time experimental' from the sentence. Also change 'Thus the interpretation of their results has certain limitations' to 'Thus, the statistical representation of their results is limited'.
Line 70: change 'climatic conditions' to 'general climate'.
Line 72: change 'beginning in 2013 has deployed routine sounding systems at Shiquanhe, Gaize, and Shenzha stations of the western TP (Fig. 1)' to 'has made routine sounding launches at Shiquanhe, Gaize, and Shenzha stations of the western TP (Fig. 1) since 2013'.
Line 82: change 'Section 4 gives major factors…' to 'Section 4 examines major factors…'

Line 94: change 'After the quality of the sounding observational data, we finally select the periods from 15 June to 31 July 2013, from 15 June to 31 August 2014, and from 1 June to 31 August 2015 in this study' to

'After quality control of the sounding data, we selected data from three time periods for this study: 15 June to 31 July 2013, 15 June to 31 August 2014, and 1 June to 31 August 2015'.

Line 95: change 'There are a total of 11,635 sounding profiles (Fig. 1a) and 4757, 2049, and 4841 profiles separately at 08:00 BJT (Fig. 1b), 14:00 BJT (Fig. 1c), and 20:00 BJT (Fig. 1d) for 19 stations over the TP' to

'There are a total of 11,635 sounding profiles (Fig. 1a) from 19 stations over the TP region consisting of 4757, 2049, and 4841 profiles at 08:00 BJT (Fig. 1b), 14:00 BJT (Fig. 1c), and 20:00 BJT (Fig. 1d), respectively'.

Note the numbers do not add up to the total here.

Lines 97 and 101: change 'sample number' to 'sample size'.

Line 99: 'Thus we also select the operational observation records that correspond to the intensive observation records'. Unclear sentence. Did you mean you subsampled the original dataset to only take those soundings that were made at the time when soundings of the test group dataset were made?

Line 105: change 'few obstacles' to 'few vegetations'.
Line 107: change '02:00 BJT, 08:00 BJT, 14:00 BJT, and 20:00 BJT' to '02:00, 08:00, 14:00, and 20:00 BJT'.

Line 110: Needs to give more details on how the interpolation of the original sounding data were made? Any filtering or smoothing during the interpolation process?

Line 117: 'For both the CBL and NBL, the ABL height is calculated as the height at which an air parcel rising adiabatically from the surface becomes neutrally buoyant (Stull 1988)'. To be clear about what you are doing, you may want to add : 'Practically, the ABL height is the level where it potential temperature is the same as that at the lowest sounding level'.

You also need to clarify the definition of the SBL height. I believe you use the height of maximum wind in the LLJ, not the 'maximum wind shear'.

Line 123: change 'and diurnal transitions (from day to night and from night to day)' to 'and day/night transitions'.

Line 125: how do you define the 'local standard time' for this location?

Line 144: 'the ABL height continues to increase in the WTP'. Is the mean height greater than 14:00 BJT to justify the 'continues to increase'? It does not look like it in the figure.

Line 150: 'The ABL height reaches the maximum in the late afternoon.' This is not clearly seen in the data in Figure 2. You may want to change to: 'Figure 2 shows continued increase in BLH in the west-most stations from 14:00 to 20:00 BJT.'

Line 165: 'Figure 4 shows the distribution of occurrence frequency of different ABL types at 08:00 BJT, 14:00 BJT, and 20:00 BJT. It is clear that the occurrence frequency shows significant diurnal variations for the SBL and CBL'. This statement is misleading. There should not be diurnal variation of SBL and CBL. The results are simply consistent with the diurnal evolution of the ABL with prevalent CBL during the sunlight hours and SBL at night.  Similarly, the

discussion of the 'diurnal variation' of NBL should be done with caution. It also would be helpful to provide the sunrise and sunset hours at representative sites of WTP and ETP to illustrate the time difference in the CBL→SBL or SBL → CBL transition. See also my comments in the list of 'Major Concerns'.

Line 209: why are the RMSE in percentage here?

Line 216: change 'larger NBL and CBL heights' to 'deeper NBLs and CBLs'.

Line 221: change 'A lot of studies' to 'Many previous studies'.

Line 225: change '…observations at SQH, NQ, and LZ stations, analyzing the…' to '…observations at SQH, NQ, and LZ stations to analyze the…'

Line 228: Any reasons for doing the '6-hour mean' in Figure 8? Which 6-hour window did you use, or was it a running mean? Please clarify.

Line 230: remove 'That is'. Change 'which supports the…' to 'which is consistent with the…'.

Line 233: 'The maximum value of SHF is…'. You should use the mean values, not the maximums.
Line 237: 'It is clear that the peak of the SHF diurnal variation occurs earlier compared to that of the ABL height
at SQH station.' Unclear sentence. Reword.
Line 238: again, how was the LST defined?
Line 239: 'This difference in 240 SHF between SQH and LZ stations is possibly associated with more cloud cover (reducing the solar radiance at the surface)'. It is better to make your statement about cloud cover after the next paragraph, if it is true.

Line 260-265: Good discussions about the soil moisture effects on SHF. What about latent heat flux (LHF)? Unless LHF is in general small, which may be true in your case for both WTP and ETP, it is also an important forcing for the ABL. But you should at least mention latent heat in this discussion.

Line 300: change 'That is, in' to 'In'.
Line 311: 'for providing the data available', delete.

Figure 1. Larger font size are needed for axis labels, station names, as well as the number of soundings. In figure caption: change 'Some letters are for the abbreviated names of stations. The green line is for the topography above 3 km.' to 'Some station names are given as abbreviations in (a) and the green lines shows the contour of terrain height at 3 km'.

Figures 2 d,e, and f, needs larger font size for axis labels and legends.

Figure 8 caption, change 'radiation flux' to 'irradiance'.

---

## Referee Comment (RC3) · Anonymous Referee #3 · 23 Nov 2020

In the ACP manuscript "Characteristics of the summer atmospheric boundary layer height over the Tibetan Plateau and influential factors" by Che and Zhao, authors used a great set of rawinsonde launches for more than two year period to analyze the surface-forcing governing the ABL depth variability in summer time. Of course, authors did not bring up any discussion why the other seasons were missing in the manuscript. The authors have made some nice time-series analyses and presented a great set of observational findings. However, at many instances, the manuscript lacks the interpretation of the results. I will encourage to consider the following points during the revisions. • In the abstract: A big picture of the problem for the region for ABL research needs to be mentioned. • It is mentioned "The SBL accounts for 85% of

the TP ABL" and also mentioned in the very next line, "The CBL accounts for 77% of the TP ABL" needs some clarification or needs to be rephrased. Otherwise, they contradict in general sense. • "The ABL height exhibits a large west-east difference, with a mean height above 2000 m in the western TP and around 1500 m in the eastern TP." Did you refer to the daytime well-mixed CBL here? • In the numbers, authors need to mention whether this is m AGL or m MSL. Since spatial variability is mentioned underlying orography will play a role if these numbers are in MSL. Please clarify. • Line 30: "ABL height in climate prediction". Authors need to bring an appropriate reference here. There is only one study that directly refers to climate projection. Please refer to the following one: "Differences in the efficacy of climate forcings explained by variations in atmospheric boundary layer depth" • Line 41: "The ABL height can be calculated from temperature, humidity, and wind profiles (Seibert et al., 2000; Seidel et al., 2010; Davy, 2018)." Please add a reference for numerical simulation as well since researchers are using models as well for this purpose. • Line 48: "solar altitude angle with respect to latitude" Please refer to Seidel et al., 2010 • Line 69: "results has certain limitations" What are those? Please be specific here. • Line 94: Quality check • Line 108: "operational observation of total cloudiness" Are these from reanalysis or from ceilometers? • Line 150: "This west-east difference increases from noon to the late afternoon." Authors need to bring the concept of west-to-east march of the solar timing given span of 20 deg longitude would cause some "real" solar timing issue although in the region there is no time zone separations and BJT is used here. According to classical rule of "15 degrees of longitude per hour", it will result in at least local time difference of 80 minutes or little more from western site to eastern site. Thus, the increase in the west-east gradient is also attributed to some extent to this "real" local timing differences. See Seidel et al. 2010 and other relevant studies as well. • Additionally, authors need to acknowledge the above-mentioned topic in other discussion where they brought up the west-east gradient changes from noon to late afternoon. • "Figure 3a-c shows the spatial distribution of the SBL height" How did they classify SBL regime during daytime soundings? Please clarify. • "remarkable diurnal variation." This is a qualitative statement unless some other SBL regimes are referred here for the contrasting scenario since SBL variability is in general low. Did you refer to the spatial variability? Please justify the causes for this then! • Throughout the results section, authors need to bring some discussion of the causes for these findings. Otherwise, it appears as reporting of the observed variability. • For section "3.2 Characteristics of SBL, NBL, and CBL heights" I will highly recommend authors performing the analyses of ABL depth growth rates which is most appropriate parameters that they wanted to discuss mentioned in the title and the abstract. Please see the feasibility of applying estimation of daily ABL depth growth rates • Several discussion via the frequency distribution analyses for ETP/WTP, authors need to decide the aim of these analyses. The results are presented with respect to findings and results without taking care of their interpretations. • Interpretation part. Line 23: "when SHF is strong, the turbulent motion is strong and the ABL height develops" True in general. What about the lag of ABL development since a number of studies showed that even after SHF attains it's maximum daytime value, ABL depth growth continues till the time of early evening transitions. I would like to see some results in this respect between ETP and WTP and that will clearly illustrate the differences in the surface forcings the authors have tried to engage the readers.

Finally, authors should consider that some comparisons with regional scale variability of ABL depths (m MSL or m AGL , be consistent) should be presented and main conclusions why this study makes an unique contribution to the field emphasizing the new processes learned for very deep ABL over the region as reported in a number of past studies.
* * *

---

## Author Comment (AC1) · 3 Jan 2021

Dear reviewer,

We are grateful for your insightful comments that help us to improve this manuscript. We carefully address issues in your comments. Please see below our point-to-point responses to your specific comments.

**General Comments**

Question 1: It is necessary and indispensable that the manuscript contains a detailed description of the formation and evolution of the planetary boundary layer.

Response: Based on your comments, we have concluded a detailed description of the formation and evolution of the planetary boundary layer in lines 41-61.

Question 2: The equation for the potential temperature difference (PTD) on line 115 is very vague and poorly understood. Authors should make a greater effort to characterize the physical criteria that allow choosing and safety to identify the types of the atmospheric boundary layer. It is important to consider that buoyancy effects make the convective and stable ABLs strikingly distinct.

Responses: Firstly, according to your suggestions, we have added statements to characterize the physical criteria that allow choosing and safety to identify the atmospheric boundary layer height. Secondly, we have added some explanations for the physical criteria of the PTD method to identify the types of the atmospheric boundary layer. The PTD method identifies the stable and convective boundary layers by judging the stability of the near surface layer atmosphere considering that buoyancy effects make the convective and stable ABLs strikingly distinct. Thirdly, we have added a detailed procedure for calculating the ABL height, the illustration of idealized atmospheric boundary layer (ABL) regimes and ABL height determination procedure, and some examples of the derived potential temperature (PT) profiles from soundings for the three types of ABL. The associated statements are in lines 137-172 and Fig. 2.

Question 3: the authors should consider in their analysis the fact that "The neutral ABL is rare because small virtual temperature differences in the ABL can cause large buoyancy patterns". How the authors identify this particular type of ABL? The authors also need to build vertical temperature and wind profiles and display them in the study.

Response: We have added the discussion for the physical criteria of the PTD method to identify the NBL and provided some examples of the derived potential temperature (PT) profiles from soundings for the three types of ABL (in lines 137-172 and Fig. 2).

Major comments
1. Line 115: The PTD classification is a fundamental criterium for the present manuscript. As a consequence, the authors must provide a more detailed discussion of the employed methodology to obtain the heights of the distinct ABL types. As the manuscripts basically observational data analysis, is not enough for the readers the citations presented.

Response: Following your suggestion, we have added a detailed description of the employed methodology to obtain the heights of the distinct ABL types and provided some examples of the derived potential temperature (PT) profiles from soundings for the three types of ABL. The associated statements are seen in lines 137-172 and Fig. 2.

2. Line 155: How a SBL can occur at noon (14:00 BJT). In this daytime period, there is a CBL. How the CBL height is near to the NBL height? The authors need to clarify.

Response: Our result shows that the SBL mainly occurs in the early morning, while the CBL mainly occurs at noon and in the late afternoon. The NBL does not show a remarkable diurnal variation. Nevertheless, the daytime SBL and the night-time CBL may also occur with low frequencies in the TP, which is likely due to the 'abnormal' forcing associated with certain synoptic conditions or cloud coverage (Medeiros et al., 2005; Poulos et al., 2002; Stull, 1988). See lines 230-235.

Stull (1998) and Blay-Carreras et al. (2014) revealed that the NBL often occurs in the transition periods between the CBL and the SBL. Since these transitions occur rapidly, the NBL may have the same characteristics in the state variables as the CBL prior to the transition although the dynamic forcing in the NBL maybe weaker compared to the CBL. Our result also shows that the CBL and NBL heights display the similar character. This result is consistent with those from Zhang et al (2017). See lines 254-258.

In addition, the similarity between the CBL and NBL may also be related to the ABL type identification scheme. The neutral stratification condition ( $\sigma = 0$ ) is rare in nature. In our calculation, the threshold value of the NBL is set to -1.0 to 1.0, which is consistent

**ACPD**
with Liu and Liang (2010). Consequently, some SBLs and CBLs with weak stratification will be identified as NBLs. See lines 150-152.

Minor comments

1. abstract "The SBL accounts for 85% of the TP ABL. At noon, there is a wide distribution in the ABL height up to 4000 m. The CBL accounts for 77% of the TP ABL, with more than 50% of the CBL height above 1900 m." Please rewrite more clearly this statement. For this reviewer the above statistics are confused.

Response: Thanks. Indeed, our statements should add the time frame to avoid any possible misunderstanding. According to your comments, we have changed (in lines 19-23).

2. Line 24: The authors need to present a better definition of the ABL.

Response: Based on your comments, we have modified the definition of the ABL (in lines 31-35).

3. Line 154: Please correct the hour "00:80 BJT"

Response: We have corrected (in line 224).

**ACPD**

---

## Author Comment (AC2) · 3 Jan 2021

Dear reviewer,

We are grateful for your insightful comments that help us to improve this manuscript. We carefully address issues in your comments. Please see below our point-to-point responses to your specific comments.

Main concerns

1. Although the presentation of the manuscript has good logic flow, descriptions of the data processing or the results can be confusing at places. The clarity of the manuscript

can be improved. I have made some specific suggestions listed in the minor concerns, but the authors should go through the manuscript very carefully or get help from people experienced in writing scientific articles in English.

Response: According to your suggestions, we have modified the text listed in your minor concerns and carefully revised the language.

2. The manuscript can be enhanced if the methodology in defining the ABL types using the PTD are revisited in the discussion section. For the SBL, the mode of the ABL height is around 300 m, suggesting that the PTD represents the temperature gradient in the main body of the stable ABL. For the CBL, since few measurements shows CBL height less than 500 m (except at 20:00 BJT), the 50 m height is likely within the surface layer. The 250 m level, on the other hand, can be in the surface layer or in the well-mixed portion of the CBL depending on the CBL height (assuming the surface layer is ∼10% of the ABL). The PTD in this case represent approximately the potential temperature difference in the surface layer or between the surface and the well-mixed CBL. The meaning of the PTD for the NBL should be similar to that in the CBL except with a smaller temperature difference. Clarifications like this should be helpful to the readers. Also, how sensitive are the results to the choice of ðÌIJŐ? My general feeling is that their results are not sensitive to the choice of ðÌIJŐ since the results of the CBL and NBL are very similar. However, the authors should make appropriate comments on the sensitivity issue.

Response: Firstly, according to your suggestions, we have added statements to characterize the physical criteria that allow choosing and safety to identify the atmospheric boundary layer height. Secondly, we have added some explanations for the physical criteria of the PTD method to identify the types of the atmospheric boundary layer. Thirdly, we have added a detailed procedure for calculating the ABL height, the illustration of idealized atmospheric boundary layer (ABL) regimes and ABL height determination procedure, and some examples of the derived potential temperature (PT) profiles from soundings for the three types of ABL. The associated statements are in lines

137-172 and Fig. 2.

Meanwhile, according to your suggestions, we have added the appropriate comments on the sensitivity issue of the ðÌÌJÕ in lines 150-152 and 165-166.

3. The overall results in this manuscript is consistent with the diurnal evolution of the ABL with the daytime deep CBLs and night-time shallow SBL. There are also occurrences of daytime SBLs and night-time CBLs although the frequencies of occurrence for both are small. The daytime SBL or night-time CBL are likely results of 'abnormal' forcing associated with certain synoptic conditions or cloud coverage. The authors mentioned a few times throughout the manuscript about the 'diurnal variations' of the SBL or the CBL (e.g., Lines 276, 290). These wordings are misleading and should be revised. It would be interesting to look into the mechanisms of the occurrence of daytime SBL and night-time CBL, but it may be beyond the scope of this paper.

Response: Thank for your comments. According to your suggestions, we have changed the "diurnal variations" to "temporal variations" and added this comment. The associated statements are seen in lines 230-235, 237, 241, and 377.

Minor points

1. Line 14: 'The SBL accounts for 85% of the TP ABL' should add the time frame here to avoid misunderstanding: 'The SBL observed during this time accounts for 85% of the TP ABL'

Response: We have changed in line 19.

2. Line 15: 'The ABL height exhibits. . ..', again, need to specify time: 'The ABL height at noon exhibits. . .'

Response: We have changed in line 20.

3. Line 20: 'For the western (eastern) TP. . .' , make it 'In general, for the western (eastern) TP. . .'

[Figure]

Response: We have modified the text (in line 26).

4. Line 28: change 'convective transmission' to 'convective transport'.

Response: We have changed (in line 36).

5. Line 56: change 'have addressed' to 'found'; also change 'can be as high as 2000–3000 m' to 'can reach 2000–3000 m'.

Response: We have modified the text (in line 78).

6. Line 57: change 'Song et al. (1984) examined the ABL height at Gaize station of the western TP is above 3000 m, while the ABL heights….' To 'Song et al. (1984) found the ABL height at Gaize station of the western TP to be above 3000 m, while the ABL heights….'

Response: It has been changed in lines 80-81.

7. Line 62: 'These results show that the ABL height over the TP varies greatly with position and season'. Change 'position' to 'location'.

Response: We have changed in line 80.

8. Line 67: change 'and less-developed logistics' to 'logistic challenges'.

Response: It has been changed in line 89.

9. Line 68: remove 'a short-time experimental' from the sentence. Also change 'Thus the interpretation of their results has certain limitations' to 'Thus, the statistical representation of their results is limited'.

Response: It has been changed in line 91.

10. Line 70: change 'climatic conditions' to 'general climate'.

Response: It has been changed in line 92.

11. Line 72: change 'beginning in 2013 has deployed routine sounding systems at

Shiquanhe, Gaize, and Shenzha stations of the western TP (Fig. 1)' to 'has made routine sounding launches at Shiquanhe, Gaize, and Shenzha stations of the western TP (Fig. 1) since 2013'.

Response: It has been changed in lines 94-95.

12. Line 82: change 'Section 4 gives major factors...' to 'Section 4 examines major factors...'

Response: It has been changed in line 106.

13. Line 94: change 'After the quality of the sounding observational data, we finally select the periods from 15 June to 31 July 2013, from 15 June to 31 August 2014, and from 1 June to 31 August 2015 in this study' to 'After quality control of the sounding data, we selected data from three time periods for this study: 15 June to 31 July 2013, 15 June to 31 August 2014, and 1 June to 31 August 2015'.

Response: It has been changed in lines 118-120.

14. Line 95: change 'There are a total of 11,635 sounding profiles (Fig. 1a) and 4757, 2049, and 4841 profiles separately at 08:00 BJT (Fig. 1b), 14:00 BJT (Fig. 1c), and 20:00 BJT (Fig. 1d) for 19 stations over the TP' to 'There are a total of 11,635 sounding profiles (Fig. 1a) from 19 stations over the TP region consisting of 4757, 2049, and 4841 profiles at 08:00 BJT (Fig. 1b), 14:00 BJT (Fig. 1c), and 20:00 BJT (Fig. 1d), respectively'. Note the numbers do not add up to the total here.

Response: We have modified the text in lines 120-122.

15. Lines 97 and 101: change 'sample number' to 'sample size'.

Response: It has been changed in lines 122, 123, and 127.

16. Line 99: 'Thus we also select the operational observation records that correspond to the intensive observation records'. Unclear sentence. Did you mean you subsampled the original dataset to only take those soundings that were made at the time when

soundings of the test group dataset were made?

Response: According to this comments, we have modified the text. See lines 125-127.

17. Line 105: change 'few obstacles' to 'few vegetation'.

Response: It has been changed in line 130.

18. Line 107: change '02:00 BJT, 08:00 BJT, 14:00 BJT, and 20:00 BJT' to '02:00, 08:00, 14:00, and 20:00 BJT'.

Response: It has been changed in lines 132-133.

19. Line 110: Needs to give more details on how the interpolation of the original sounding data were made? Any filtering or smoothing during the interpolation process?

Response: We have modified the text. See lines 144-146.

20. Line 117: 'For both the CBL and NBL, the ABL height is calculated as the height at which an air parcel rising adiabatically from the surface becomes neutrally buoyant (Stull 1988)'. To be clear about what you are doing, you may want to add: 'Practically, the ABL height is the level where it potential temperature is the same as that at the lowest sounding level'. You also need to clarify the definition of the SBL height. I believe you use the height of maximum wind in the LLJ, not the 'maximum wind shear'.

Response: According to your suggestion, we have added a more detailed discussion about the employed methodology to obtain the CBL height and revised the definition of the SBL height. See lines 154-169.

21. Line 123: change 'and diurnal transitions (from day to night and from night to day)' to 'and day/night transitions'.

Response: It has been changed in line 175.

22. Line 125: how do you define the 'local standard time' for this location?

Response: We use the local solar time as the LST and add this statement in line 177.

23. Line 144: 'the ABL height continues to increase in the WTP'. Is the mean height greater than 14:00 BJT to justify the 'continues to increase'? It does not look like it in the figure.

Response: We have changed in lines 200-202.

24. Line 150: 'The ABL height reaches the maximum in the late afternoon.' This is not clearly seen in the data in Figure 2. You may want to change to: 'Figure 2 shows continued increase in BLH in the west-most stations from 14:00 to 20:00 BJT.'

Response: According to your suggestion, we have modified. See lines 205-206.

25. Line 165: 'Figure 4 shows the distribution of occurrence frequency of different ABL types at 08:00 BJT, 14:00 BJT, and 20:00 BJT. It is clear that the occurrence frequency shows significant diurnal variations for the SBL and CBL'. This statement is misleading. There should not be diurnal variation of SBL and CBL. The results are simply consistent with the diurnal evolution of the ABL with prevalent CBL during the sunlight hours and SBL at night. Similarly, the discussion of the 'diurnal variation' of NBL should be done with caution. It also would be helpful to provide the sunrise and sunset hours at representative sites of WTP and ETP to illustrate the time difference in the CBL SBL or SBL CBL transition. See also my comments in the list of 'Major Concerns'.

Response: According to your suggestions, we have changed in lines 225, 229, 237, and 241. Moreover, we have added this comments in the discussions in lines 230-235.

26. Line 209: why are the RMSE in percentage here?

Response: This RMSE is for the occurrence frequency. In our revised manuscript, we have changed "with root-mean-square errors (RMSEs) between 1.1% and 2.7%" to "with root-mean-square errors (RMSEs) of the occurrence frequency between 1.1% and 2.7%" in lines 282-283.

27. Line 216: change 'larger NBL and CBL heights' to 'deeper NBLs and CBLs'.

Response: It has been changed in line 291.

28. Line 221: change 'A lot of studies' to 'Many previous studies'.

Response: It has been changed in line 295.

29. Line 225: change '. . .observations at SQH, NQ, and LZ stations, analyzing the. . .' to '. . .observations at SQH, NQ, and LZ stations to analyze the. . .'

Response: It has been changed in line 300.

30. Line 228: Any reasons for doing the '6-hour mean' in Figure 8? Which 6-hour window did you use, or was it a running mean? Please clarify.

Response: We have added the reasons for doing the '6-hour mean' and modified the text in lines 308-310.

31. Line 230: remove 'That is'. Change 'which supports the. . .' to 'which is consistent with the. . .'.

Response: It has been changed in line 312.

32. Line 233: 'The maximum value of SHF is. . .'. You should use the mean values, not the maximums.

Response: It has been changed in lines 315-316.

33. Line 237: 'It is clear that the peak of the SHF diurnal variation occurs earlier compared to that of the ABL height at SQH station.' Unclear sentence. Reword.

Response: It has been changed. See lines 320-321.

34. Line 238: again, how was the LST defined?

Response: We use the local sidereal time as the LST and add this statement in line 177.

35. Line 239: 'This difference in SHF between SQH and LZ stations is possibly asso-

ciated with more cloud cover (reducing the solar radiance at the surface)'. It is better to make your statement about cloud cover after the next paragraph, if it is true.

Response: We have deleted this content on cloud cover in lines 318-325 because it is discussed in the next paragraph.

36. Line 260-265: Good discussions about the soil moisture effects on SHF. What about latent heat flux (LHF)? Unless LHF is in general small, which may be true in your case for both WTP and ETP, it is also an important forcing for the ABL. But you should at least mention latent heat in this discussion.

Response: We have calculated the mean LHF at three stations (not included in the text). The result indicates that the kinematic moisture flux (KMF) is general small for both WTP and ETP. See lines 303-308.

37. Line 300: change 'That is, in' to 'In'.

Response: It has been changed in line 386.

38. Line 311: 'for providing the data available', delete.

Response: It has been changed in line 406.

39. Figure 1. Larger font size is needed for axis labels, station names, as well as the number of soundings. In figure caption: change 'Some letters are for the abbreviated names of stations. The green line is for the topography above 3 km.' to 'Some station names are given as abbreviations in (a) and the green lines shows the contour of terrain height at 3 km'.

Response: It has been changed in lines 571-573.

40. Figures 2 d,e, and f, needs larger font size for axis labels and legends.

Response: It has been changed in lines 583-587.

41. Figure 8 caption, change 'radiation flux' to 'irradiance'.

Response: It has been changed in line 614.

---

## Author Comment (AC3) · 3 Jan 2021

Dear reviewer,

We are grateful for your insightful comments that help us to improve this manuscript. We carefully address issues in your comments. Please see below our point-to-point responses to your specific comments.

1. Of course, authors did not bring up any discussion why the other seasons were missing in the manuscript.

Response: The TIPEX-III experiment carried out the intensive observations in the TP

region at noon (14:00 BJT) during summer, which provide a better dataset for studying the ABL during summer. In other seasons, there are no observations at 14:00 BJT. Thus summer is selected in this study. The associated statements have been added in lines 93-100 and 116-118. In addition, a statement has been added in Summary and Discussion (in lines 402-404).

2. However, at many instances, the manuscript lacks the interpretation of the results.

Response: The original manuscript was structured to show the results in Section 3 and present physical explanations in Section 4. We have realized, based on your comments, that arrangement may result in disconnections between the results and their discussions. In the revised manuscript, we have included some interpretations and the possible physical reason analyses immediately following the results. See our response to question 16 for the specific revisions.

3. In the abstract: A big picture of the problem for the region for ABL research needs to be mentioned.

Response: According to your suggestion, we have added some statements about the existing problem for the TP ABL research in lines 9-11.

4. It is mentioned "The SBL accounts for 85% of the TP ABL" and also mentioned in the very next line, "The CBL accounts for 77% of the TP ABL" needs some clarification or needs to be rephrased. Otherwise, they contradict in general sense.

Response: Thanks. Indeed, our statements should add the time frame to avoid any possible misunderstanding. According to your comments, we have changed (in lines 19-23).

5. The ABL height exhibits a large west-east difference, with a mean height above 2000 m in the western TP and around 1500 m in the eastern TP." Did you refer to the daytime well-mixed CBL here?

Response: It is due to our unclear statement. This sentence is for ABL not only for
CBL (in line 21).

6. In the numbers, authors need to mention whether this is m AGL or m MSL. Since spatial variability is mentioned underlying orography will play a role if these numbers are in MSL. Please clarify.

Response: It is m above ground level (AGL). Following your suggestion, we have changed "m" to "m AGL" in the revised manuscript.

7. Line 30: "ABL height in climate prediction". Authors need to bring an appropriate reference here. There is only one study that directly refers to climate projection. Please refer to the following one: "Differences in the efficacy of climate forcings explained by variations in atmospheric boundary layer depth"

Response: We have added the associated references on the ABL height in climate prediction (in lines 37-38).

References:

Garratt, J. R., 1993: Sensitivity of climate simulations to land-surface and atmospheric boundary-layer treatments—A review. J. Climate, 6, 419–448.

Esau, I., and S. Zilitinkevich, 2010: On the role of the planetary boundary layer depth in the climate system. Adv. Sci. Res., 4, 63–69.

Davy, R., and I. Esau, 2016: Differences in the efficacy of climate forcings explained by variations in atmospheric boundary layer depth. Nat. Commun., 7, 11690.

8. Line 41: "The ABL height can be calculated from temperature, humidity, and wind profiles (Seibert et al., 2000; Seidel etal., 2010; Davy, 2018)." Please add a reference for numerical simulation as well since researchers are using models as well for this purpose.

Response: We have added the following references for numerical simulation (in lines 63-64).

**ACPD**
References:

Holtslag, B. and B. A. Boville, 1993: Local versus nonlocal boundary-layer diffusion in a global climate model. J. Climate, 6, 1825–1842.

Bosveld, F. C., and Coauthors, 2014b: The third GABLS intercomparison case for evaluation studies of boundary-layer models. Part B: Results and process understanding. Bound-Layer Meteor., 152, 157–187.

9. Line 48: "solar altitude angle with respect to latitude" Please refer to Seidel et al., 2010.

Response: We have changed (in lines 68-70).

10. Line 69: "result has certain limitations" What are those? Please be specific here.

Response: According to another reviewer's suggestion, we have changed "Thus, the statistical representation of their results is limited" (in line 91).

11. Line 94: Quality check

Response: It is due to our mistake. We have changed to "quality control" (in line 119).

12. Line 108: "operational observation of total cloudiness" Are these from reanalysis or from ceilometers?

Response: It is the manual ground-based cloud cover observations from the China Meteorological Administration, and has been used to analyze the relationship between the ABL height and cloud cover in China by Guo et al. (2016) and Zhang et al. (2017). The associated statements are seen in lines 132-134.

References:

Guo, J. P., Miao, Y. C., Zhang, Y., Liu, H., Li, Z. Q., Zhang, W. C., He, J., Lou, M. Y., Yan, Y., Bian, L. G., Zhai, P. M.: The climatology of planetary boundary layer height in China derived from radiosonde and reanalysis data, Atmospheric. Chemistry. and
Physics., 16, 13309-13319, doi:10.5194/acp-16-13309-2016, 2016.

Zhang, W., Guo, J., Miao, Y., Liu, H., Yang, S., Fang, Z., He, J., Lou, M. Y., Yan, Y., Li, Y., Zhai, P. M.: On the summertime planetary boundary layer with different thermodynamic stability in China: a radiosonde perspective, Journal of Climate., 31, doi: 10.1175/jcli-d-17-0231.1, 2017.

13. Line 150: "This west-east difference increases from noon to the late afternoon." Authors need to bring the concept of west-to-east march of the solar timing given span of 20 deg longitude would cause some "real" solar timing issue although in the region there is no time zone separations and BJT is used here. According to classical rule of "15 degrees of longitude per hour", it will result in at least local time difference of 80 minutes or little more from western site to eastern site. Thus, the increase in the west-east gradient is also attributed to some extent to this "real" local timing differences. See Seidel et al. 2010 and other relevant studies as well. Additionally, authors need to acknowledge the above-mentioned topic in other discussion where they brought up the west-east gradient changes from noon to late afternoon.

Response: We agree with you that the increase in the west-east gradient of ABLH (including CBL height and NBL height) from noon to the late afternoon is also attributed to some extent to the "real" local timing differences. According to Seidel et al. (2010, 2012) and Guo et al. (2016), in the revised manuscript, we have added an explanation for the increasing west-east difference of the ABLH over the TP from noon to the late afternoon (in lines 205-210).

In addition, the phenomenon of "the SBL/CBL mainly occurring in the ETP/WTP at 20:00 BJT" is also related to the above-mentioned topic. In the revised manuscript, we have added an explanation, that is, the above results are consistent with the diurnal development of the ABL structure including the SBL in the early morning, the CBL at noon, and different types of ABLs between the eastern and western TP in the late afternoon because of the latitudinal difference and the resultant difference in local solar
times. Note that the observations were made simultaneously for all stations. The associated statements are seen in lines 230-233.

14. Figure 3a-c shows the spatial distribution of the SBL height" How did they classify SBL regime during daytime soundings? Please clarify.

Response: The definition of the daytime SBL height is the same as that of the nighttime SBL in Section 2.2. In this revision, we have given the identification method of SBL in detail, the corresponding diagram of SBL, and an example (in lines 137-150, 167-172, and Fig. 2).

15. "remarkable diurnal variation." This is a qualitative statement unless some other SBL regimes are referred here for the contrasting scenario since SBL variability is in general low. Did you refer to the spatial variability? Please justify the causes for this then!

Response: We agree with you that the SBL variability is in general low, which is consistent with our results. We have changed in lines 261-263.

16. Throughout the results section, authors need to bring some discussion of the causes for these findings. Otherwise, it appears as reporting of the observed variability.

Response: Following your suggestion, we have brought some discussions of the causes in the third section. A detailed physical discussion has been given in the fourth section. The detailed statements are as follows. We have added an explanation for the regional difference of the ABLH over the TP at 14:00 BJT (in lines 196-198), an explanation for the increasing west-east difference of the ABLH over the TP from noon to the late afternoon (in lines 206-210), an explanation for the results of spatial and temporal distribution of occurrence frequency of different types of ABL (in lines 230-235), and a discussion for the results of temporal variations of all types of ABLH over the TP (in lines 254-258).

17. For section "3.2 Characteristics of SBL, NBL, and CBL heights" I will highly rec-

**ACPD**
ommend authors performing the analyses of ABL depth growth rates which is most appropriate parameters that they wanted to discuss mentioned in the title and the abstract. Please see the feasibility of applying estimation of daily ABL depth growth rates.

Response: According to your suggestion, we have added the growth rates of the ABLH from 08:00BJT to 14:00 BJT and from 14:00 BJT to 20:00 BJT, and added the associated statements (in lines 212-220) and Fig. 4.

18. Several discussion via the frequency distribution analyses for ETP/WTP, authors need to decide the aim of these analyses. The results are presented with respect to findings and results without taking care of their interpretations.

Response: The analyses of the frequency distribution of ABLH (Fig. 6g, h) intend to show the differences between the two regions of the TP from a different angle than the differences in the mean ABLH shown in Fig. 7. This discussion focused on the most frequently observed ABLH and how these most common ABLH varied gradually from EPT to WPT. It is true that we have not included much physical interpretation here. However, discussions are added later when the boundary layer forcing factors are discussed in section 4. The text has been modified to make our objectives of the discussion clear (in lines 271-277).

19. Interpretation part. Line 23: "when SHF is strong, the turbulent motion is strong and the ABL height develops" True in general. What about the lag of ABL development since a number of studies showed that even after SHF attains it's maximum daytime value, ABL depth growth continues till the time of early evening transitions. I would like to see some results in this respect between ETP and WTP and that will clearly illustrate the differences in the surface forcings the authors have tried to engage the readers.

Response: Thanks for your comments. Following your opinions, we have added the analysis of diurnal variations of SHF and ABLH. The associated statements are seen in lines 318-325.

**ACPD**
20. Finally, authors should consider that some comparisons with regional scale variability of ABL depths (m MSL or m AGL, be consistent) should be presented and main conclusions why this study makes an unique contribution to the field emphasizing the new processes learned for very deep ABL over the region as reported in a number of past studies.

Response: As we understand for this question, we have compared the regional variability of the ABLH in TP with the other regional variability of the ABLH such as in the United States (Seidel et al., 2012) and in China (Guo et al., 2016) in the discussion (in lines 370-375 and 395-400), and also have explicitly stated the unique contribution of this study to the ABL in the TP (in lines 393-395).

**ACPD**

---

## Author Response (AR2)

**Response to the reviewers' comments**

**Response to the comments of Reviewer #2**

**1. Line 21. Still Not clear. The ABL height exhibits a large west-east difference. Is it during daytime or nighttime or the daily average. please specify. Please see my comments in the last version as well on this.**

  **Response:** It is for the total ABL height at 14:00 BJT. We have changed to "there are a wide distribution in the ABL height up to 4000 m AGL and a large west-east difference for the total ABL height at noon (14:00 BJT), with a mean height above 2000 m AGL in the western TP and around 1500 m AGL in the eastern TP" in lines 20-22.

**2. My previous comment on Line 150, about west-east differences have not been addressed properly. It will be important if the authors could bring previous studies on this topic in the region and bring more clarity in the discussion. Please see figure 3 in Lee and Pal (2017, On the potential of 25 years (1991-2015) of rawinsonde measurements for elucidating key climatological and spatiotemporal patterns of afternoon boundary layer depths over the contiguous US) for an extended discussion on this topic. If the authors could bring such results from the region reported here, that would do.**

  **Responses:** Based on your suggestions, we have added the reference of Lee and Pal (2017). According to the Figure 3 in Lee and Pal (2017), we have illustrated the different time zones in the study region in the revised Fig. 1b-c in line 588. During the evening transition, the daytime boundary layer undergoes a transition to the night-time boundary layer. Since the TP spans almost 1.5 time zones from west to east (Fig. 1c), the local solar time is earlier in the west than in the east. At 20:00 BJT, it is at 17:20 LST for the westernmost SQH station and is 18:50 LST for the easternmost HY station, with a

difference of 1.5 h. This result supports an earlier transition from the daytime ABL to the night-time ABL in the east (Seidel et al., 2010, 2012; Guo et al., 2016; Lee and Pal, 2017). Meanwhile, we also noted that this difference of the local time from east to west (2000 km) is less over the TP than over China (4000 km) (Guo et al., 2016) and the United States (4000 km) (Seidel et al., 2010, 2012; Lee and Pal, 2017). Thus the contribution of the time zone difference (1.5 h) to the regional difference of ABLH is relatively smaller in TP. The above discussions have been added in lines 211-218.

**3. Also, see some other differences and error sources discussed in Lee and Pal (2020, The Impact of Height-independent Errors in State Variables on the Determination of the Daytime Atmospheric Boundary Layer Depth using the Bulk Richardson Approach).**

 **Response:** Thanks for your comments. We have added the reference of Lee and Pal (2021), and have added the discussions about the ABLH error estimates in lines 166-172.

**Response to the comments of Reviewer #3**

**Technical corrections:**

**1. line 141, remove 'ground'.**

    **Response:** According to your comments, we have corrected (in line 142).

**2. line 190, reword 'with the well developing of the daytime ABL' to 'with the well-developed daytime ABL.**

    **Responses:** We have changed (in line 195).

**3. line 200, change 'begins to turn to the night-time feature' to 'begins to show the nighttime features.**

    **Response:** We have changed (in line 205).

**4. line 286, change 'little' to 'small'.**

    **Response:** We have changed (in line 294).

[revised manuscript text omitted]